# The First Optimal Acceleration of High-Order Methods in Smooth Convex Optimization

**Dmitry Kovalev**
KAUST[*]
dakovalev1@gmail.com

**Alexander Gasnikov**
MIPT,[†] IITP RAS,[‡] HSE[§]
gasnikov@yandex.ru

## Abstract

In this paper, we study the fundamental open question of finding the optimal high-order algorithm for solving smooth convex minimization problems. Arjevani et al. (2019) established the lower bound $\Omega\left(\epsilon^{-2/(3p+1)}\right)$ on the number of the $p$-th order oracle calls required by an algorithm to find an $\epsilon$-accurate solution to the problem, where the $p$-th order oracle stands for the computation of the objective function value and the derivatives up to the order $p$. However, the existing state-of-the-art high-order methods of Gasnikov et al. (2019b); Bubeck et al. (2019); Jiang et al. (2019) achieve the oracle complexity $\mathcal{O}\left(\epsilon^{-2/(3p+1)}\log(1/\epsilon)\right)$, which does not match the lower bound. The reason for this is that these algorithms require performing a complex binary search procedure, which makes them neither optimal nor practical. We fix this fundamental issue by providing the first algorithm with $\mathcal{O}\left(\epsilon^{-2/(3p+1)}\right)$ $p$-th order oracle complexity.

## 1 Introduction

Let $\mathbb{R}^d$ be a finite-dimensional Euclidean space and let $f(x)\colon \mathbb{R}^d \to \mathbb{R}$ be a convex, $p$ times continuously differentiable function with $L_p$-Lipschitz $p$-th order derivatives. Our goal is to solve the following convex minimization problem:

$$f^* = \min_{x \in \mathbb{R}^d} f(x). \tag{1}$$

In this work, we assume access to the $p$-th order oracle associated with function $f(x)$. That is, given an arbitrary point $x \in \mathbb{R}^d$, we can compute the function value and the derivatives of function $f(x)$ up to order $p$.

**First-order methods.** When $p = 1$, first-order methods, such as gradient descent, are typically used for solving problem (1). The lower bound $\Omega(\epsilon^{-1/2})$ on the number of the gradient evaluations required by these algorithms to find an $\epsilon$-accurate solution was established by Nemirovskij and Yudin (1983); Nesterov (2003), while the optimal algorithm matching this lower bound is called Accelerated Gradient Descent and was developed by Nesterov (1983).

**Second-order methods.** In contrast to the first-order methods, the understanding of the second-order methods ($p = 2$) was developed relatively recently. Nesterov and Polyak (2006) developed the cubic regularized variant of Newton's method. This algorithm achieves global convergence with

---

[*]King Abdullah University of Science and Technology, Thuwal, Saudi Arabia

[†]Moscow Institute of Physics and Technology, Dolgoprudny, Russia

[‡]Institute for Information Transmission Problems RAS, Moscow, Russia

[§]National Research University Higher School of Economics, Moscow, Russia

36th Conference on Neural Information Processing Systems (NeurIPS 2022).

Table 1: Comparison of the first-order, second-order and high-order methods for smooth convex optimization in the oracle complexities (see Definition 3), which depend on the smoothness constant $L_p$ (see Assumption 1), the distance to the solution $R$ (see Assumption 2), and the accuracy $\epsilon$ (see Definition 1).

| Algorithm Reference | Oracle Complexity | Order |
|---|---|---|
| Nesterov (1983) | $\mathcal{O}\left(\left(L_1 R^2/\epsilon\right)^{1/2}\right)$ | **First-Order Methods** $(p = 1)$ |
| Lower Bound (Nemirovskij and Yudin, 1983) | $\Omega\left(\left(L_1 R^2/\epsilon\right)^{1/2}\right)$ | |
| Nesterov and Polyak (2006) | $\mathcal{O}\left(\left(L_2 R^3/\epsilon\right)^{1/2}\right)$ | **Second-Order Methods** $(p = 2)$ |
| Nesterov (2008) | $\mathcal{O}\left(\left(L_2 R^3/\epsilon\right)^{1/3}\right)$ | |
| Monteiro and Svaiter (2013) | $\mathcal{O}\left(\left(L_2 R^3/\epsilon\right)^{2/7} \log(1/\epsilon)\right)$ | |
| **Algorithm 4 (This Paper)** | $\mathcal{O}\left(\left(L_2 R^3/\epsilon\right)^{2/7}\right)$ | |
| Lower Bound (Arjevani et al., 2019) | $\Omega\left(\left(L_2 R^3/\epsilon\right)^{2/7}\right)$ | |
| Nesterov (2021a) | $\mathcal{O}\left(\left(L_p R^{p+1}/\epsilon\right)^{1/p}\right)$ | **High-Order Methods** $(p \geq 2)$ |
| Nesterov (2021a) | $\mathcal{O}\left(\left(L_p R^{p+1}/\epsilon\right)^{1/(p+1)}\right)$ | |
| Gasnikov et al. (2019b) | $\mathcal{O}\left(\left(L_p R^{p+1}/\epsilon\right)^{2/(3p+1)} \log(1/\epsilon)\right)$ | |
| **Algorithm 4 (This Paper)** | $\mathcal{O}\left(\left(L_p R^{p+1}/\epsilon\right)^{2/(3p+1)}\right)$ | |
| Lower Bound (Arjevani et al., 2019) | $\Omega\left(\left(L_p R^{p+1}/\epsilon\right)^{2/(3p+1)}\right)$ | |

the oracle complexity $\mathcal{O}(\epsilon^{-1/2})$, which cannot be achieved with the standard Newton's method. Nesterov (2008) also developed an accelerated version of the cubic regularized Newton's method with $\mathcal{O}\left(\epsilon^{-1/3}\right)$ second-order oracle complexity. A few years later, Monteiro and Svaiter (2013) developed the Accelerated Hybrid Proximal Extragradient (A-HPE) framework and combined it with a trust region Newton-type method. The resulting algorithm, called Accelerated Newton Proximal Extragradient (A-NPE), achieved the second-order oracle complexity of $\mathcal{O}\left(\epsilon^{-2/7} \log(1/\epsilon)\right)$. In 2018, Arjevani et al. (2019) established the lower bound $\Omega\left(\epsilon^{-2/7}\right)$ on the number of the second-order oracle calls required by an algorithm to find an $\epsilon$-accurate solution[5], which is almost achieved by the A-NPE algorithm of Monteiro and Svaiter (2013), up to the logarithmic factor $\log(1/\epsilon)$. However, the optimal second-order algorithms for solving smooth convex minimization problems remain to be unknown.

**High-order methods.** In the case when $p > 2$, the situation is very similar to the second-order case. Nesterov (2021a) developed the generalization of the cubic regularized Newton method to the high-order case and called them tensor methods. Nesterov (2021a) provided both non-accelerated and accelerated $p$-th order tensor methods with the oracle complexity $\mathcal{O}\left(\epsilon^{-1/p}\right)$ and $\mathcal{O}\left(\epsilon^{-1/(p+1)}\right)$, respectively.[6] Later, three independent groups of researchers (Gasnikov et al., 2019a; Bubeck et al., 2019; Jiang et al., 2019) used the A-HPE framework to develop the near-optimal tensor methods with the oracle complexity $\mathcal{O}\left(\epsilon^{-2/(3p+1)} \log(1/\epsilon)\right)$. Similarly to the case $p = 2$, these algorithms match the lower complexity bound $\Omega\left(\epsilon^{-2/(3p+1)}\right)$ of Arjevani et al. (2019), up to the logarithmic factor $\log(1/\epsilon)$.

---

[5]There is also a work of Agarwal and Hazan (2018), which provides the lower complexity bounds for high-order optimization. However, their lower bounds are worse than the lower bounds of Arjevani et al. (2019).

[6]Nesterov (2021a) also provided the lower complexity bounds that coincide with the lower bounds of Arjevani et al. (2019).

## 1.1 Main Contribution: Optimal Second-Order and High Order Methods

The review of the second-order and high-order methods that we made above identifies the following fundamental open question:

*Can we design an optimal p-th order algorithm ($p \geq 2$) for solving smooth convex minimization problems with the oracle complexity matching the lower bounds?*

The lack of an answer to this question reveals a significant gap in the understanding of the high-order optimization compared to the first-order optimization. We give a positive answer to this question. That is, we provide the first optimal high-order algorithm with the $p$-th order oracle complexity $\mathcal{O}\left(\epsilon^{-2/(3p+1)}\right)$ that matches the lower bounds of Arjevani et al. (2019). This is the main contribution of our work.

## 1.2 Related Work

High-order and second-order methods have attracted a lot of interest recently. Relevant works include but are not limited to superfast second-order methods (Nesterov, 2021c), second-order methods with gradient regularization (Mishchenko, 2021; Doikov et al., 2022; Doikov and Nesterov, 2021), high-order methods for non-smooth optimization via smoothing technique (Bullins, 2020), and ball-constrained optimization Carmon et al. (2020, 2021). A more detailed review of recent advances in high-order optimization can be found in the work of Kamzolov et al. (2022).

**High-order methods for variational inequalities.** Monteiro and Svaiter (2012) also developed second-order methods for solving monotone variational inequalities and inclusions problems for operators with Lipschitz continuous derivatives, which were generalized to the high-order setting by Bullins and Lai (2022); Jiang and Mokhtari (2022). However, similarly to the near-optimal high-order methods for minimization problems (Monteiro and Svaiter, 2013; Gasnikov et al., 2019b), these algorithms have additional logarithmic factors in the complexity that appear due to the requirement of performing the binary search procedure.

Recently, Lin et al. (2022); Adil et al. (2022) removed the extra logarithmic factors and provided high-order methods for solving monotone variational inequalities and inclusions problems without any binary search procedures. Moreover, Lin et al. (2022) established lower complexity bounds that matched the proposed algorithms. Hence, the problem of finding optimal high-order algorithms for solving variational inequalities and inclusions problems is solved.

In our paper, we solve a similar problem of finding optimal high-order methods for solving smooth convex minimization problems. Note that this problem is much more challenging because optimal algorithms for solving variational inequalities and inclusion problems are typically much simpler and do not require utilising acceleration techniques.

**Concurrent work of Carmon et al. (2022).** The question of finding optimal high-order methods for solving smooth convex minimization problems was solved recently in the concurrent work of Carmon et al. (2022). However, their approach is substantially different from the approach used in this work, and their paper appeared on arXiv 11 days later than ours.

## 1.3 Paper Organization

Our paper is organized as follows: **(i)** in Section 2, we briefly introduce the tensor approximations and provide necessary assumptions and definitions; **(ii)** in Section 3, we describe the existing near-optimal high-order methods and identify their main flaws that prevent them from being optimal and practical algorithms; **(iii)** in Section 4, we describe the development of our optimal high-order algorithm and provide its theoretical convergence analysis.

## 2 Preliminaries

By $\| \cdot \| \colon \mathbb{R}^d \to \mathbb{R}$ and $\langle \cdot, \cdot \rangle \colon \mathbb{R}^d \times \mathbb{R}^d \to \mathbb{R}$, we denote the standard Euclidean norm and scalar product on $\mathbb{R}^d$. Given a $p$ times continuously differentiable function $g(x) \colon R^d \to \mathbb{R}$ and index

$i \in \{1, 2, \ldots, p\}$, by $\nabla^i g(x)[h]^i \colon \mathbb{R}^d \to \mathbb{R}$ we denote the following homogeneous polynomial:

$$\nabla^i g(x)[h]^i = \sum_{j_1, \ldots, j_i = 1}^{d} \frac{\partial^i g}{\partial x_{j_1} \cdots \partial x_{j_i}}(x) \cdot h_{j_1} \cdots h_{j_i}, \tag{2}$$

where $x = (x_1, \ldots, x_d) \in \mathbb{R}^d$, $h = (h_1, \ldots, h_d) \in \mathbb{R}^d$, and

$$\frac{\partial^i g}{\partial x_{j_1} \cdots \partial x_{j_i}}(x) \tag{3}$$

is the $i$-th order partial derivative of function $g(x)$ at point $x$ with respect to variables $x_{j_1}, \ldots, x_{j_i}$. For instance, if $i = 1$, then $\nabla^1 g(x)[h] = \langle \nabla g(x), h \rangle$, where $\nabla g(x) \in \mathbb{R}^d$ is the gradient of function $g(x)$; if $i = 2$, then $\nabla^2 g(x)[h] = \langle \nabla^2 f(x) h, h \rangle$, where $\nabla^2 f(x) \in \mathbb{R}^{d \times d}$ is the Hessian of function $f(x)$. We can write the $p$-th order Taylor approximation of function $g(x)$ at point $z \in \mathbb{R}^d$:

$$\Phi_g^p(x; z) = g(z) + \sum_{i=1}^{p} \frac{1}{i!} \nabla^i g(z)[x - z], \tag{4}$$

It is well known that the Taylor polynomial $\Phi_g^p(x; z)$ approximates function $g(x)$, if point $x$ is close enough to point $z$:

$$g(x) = \Phi_g(x; z) + R_g^p(x; z)\|x - z\|^p, \tag{5}$$

where $R_g^p(\cdot; z) \colon \mathbb{R}^d \to \mathbb{R}$ is a function that satisfies $\lim_{x \to z} R_g^p(x; z) = 0$.

As mentioned earlier, we assume that the objective function $f(x)$ of the main problem (1) is $p$ times continuously differentiable and has $L_p$-Lipschitz $p$-th order derivatives. It is formalized via the following definition.

**Assumption 1.** *Function $f(x)$ is $p$-times continuously differentiable, convex, and has $L_p$-Lipschitz $p$-th order derivatives, i.e., for all $x_1, x_2 \in \mathbb{R}^d$ the following inequality holds:*

$$\max\{|\nabla^p f(x_1)[h] - \nabla^p f(x_2)[h]| : h \in \mathbb{R}^d, \|h\| \le 1\} \le L_p \|x_1 - x_2\|.$$

Theorem 1 of Nesterov (2021a) implies that under Assumption 1, function $f(x)$ has the following convex upper bound:

$$f(x) \le \Phi_f^p(x; z) + \frac{pM}{(p+1)!}\|x - z\|^{p+1}, \tag{6}$$

where $M \ge L_p$ and $z \in \mathbb{R}^d$. Hence, an obvious approach to solving problem (1) is to perform the minimization of this upper bound instead of minimizing the function $f(x)$. This approach naturally leads to the following iterative process:

$$x^{k+1} \in \underset{x \in \mathbb{R}^d}{\text{Arg min}}\ \Phi_f^p(x; x^k) + \frac{pM}{(p+1)!}\|x - x^k\|^{p+1}. \tag{7}$$

In the case $p = 2$, this iterative process is known as the cubic regularized Newton's method of Nesterov and Polyak (2006), and in the case $p > 2$, it is known as the tensor method of Nesterov (2021a). Minimization procedures similar to (7) are widely used in high-order optimization methods. It will also be used in the development of our optimal algorithm.

We also have the following assumption which requires problem (1) to have at least a single solution $x^* \in \mathbb{R}^d$. It is a standard assumption for the majority of works on convex optimization.

**Assumption 2.** *There exists a constant $R > 0$ and at least a single solution $x^*$ to problem (1), such that $\|x^0 - x^*\| \le R$, where $x^0 \in \mathbb{R}^d$ is the starting point that we use as an input for a given algorithm for solving the problem.*

Finally, we have the following definitions that formalize the notions of $\epsilon$-accurate solution of a problem, $p$-th order oracle call, and oracle complexity of an algorithm.

**Definition 1.** *We call vector $\hat{x} \in \mathbb{R}^d$ an $\epsilon$-accurate solution of problem (1), if for a given accuracy $\epsilon > 0$ it satisfies $f(\hat{x}) - f^* \le \epsilon$.*

**Definition 2.** *Given an arbitrary vector $x \in \mathbb{R}^d$ by the $p$-th order oracle call at $x$, we denote the computation of the function value $f(x)$ and the derivatives $\nabla^1 f(x)[\cdot], \ldots, \nabla^p f(x)[\cdot]$.*

**Definition 3.** *By the $p$-th order oracle complexity of a $p$-th order algorithm for solving problem (1), we denote the number of $p$-th order oracle calls required by the algorithm to find an $\epsilon$-accurate solution of the problem for a given accuracy $\epsilon > 0$.*

---
**Algorithm 1** A-HPE Framework
---
1: **input:** $x^0 = x_f^0 \in \mathbb{R}^d$
2: **parameters:** $\sigma \in [0, 1]$, $K \in \{1, 2, \ldots\}$
3: $\beta_{-1} = 0$
4: **for** $k = 0, 1, 2, \ldots, K - 1$ **do**
5:     compute $x_f^{k+1} \in \mathbb{R}^d$, $\lambda_k > 0$ such that

$$\|\nabla f(x_f^{k+1}) + \lambda_k^{-1}(x_f^{k+1} - x_g^k)\| \leq \sigma \lambda_k^{-1} \|x_f^{k+1} - x_g^k\|, \tag{8}$$

   where $x_g^k \in \mathbb{R}^d$ and $\alpha_k \in (0, 1]$ are defined as

$$x_g^k = \alpha_k x^k + (1 - \alpha_k) x_f^k, \quad \alpha_k = \eta_k / \beta_k, \tag{9}$$

   and $\eta_k > 0$ and $\beta_k > 0$ are defined by the following system:

$$\beta_{k-1} + \eta_k = \beta_k, \quad \beta_k \lambda_k = \eta_k^2. \tag{10}$$

6:     $x^{k+1} = x^k - \eta_k \nabla f(x_f^{k+1})$
7: **end for**
8: **output:** $x_f^K$
---

# 3 Near-Optimal Tensor Methods

In this section, we revisit the state-of-the-art high-order optimization algorithms that include the A-NPE method of Monteiro and Svaiter (2013) in the $p = 2$ case and the near-optimal tensor methods of Gasnikov et al. (2019a); Bubeck et al. (2019); Jiang et al. (2019) in the general $p > 2$ case. We start with describing the key ideas behind the development of these algorithms to understand how they work. Then, we identify the main flaws of the algorithms that prevent them from being optimal and practical.

Note that the A-NPE method and near-optimal tensor methods have the following substantial similarities: **(i)** both the A-NPE and near-optimal tensor methods are based on the A-HPE framework of Monteiro and Svaiter (2013); **(ii)** the oracle complexity of the near-optimal tensor methods recovers the oracle complexity of the A-NPE method in the case $p = 2$; **(iii)** these algorithms have the same issue: the requirement to perform the complex binary search procedure at each iteration which makes them neither optimal nor practical. Hence, we will further leave out the description of the A-NPE method of Monteiro and Svaiter (2013) and consider only the near-optimal tensor methods of Gasnikov et al. (2019a); Bubeck et al. (2019); Jiang et al. (2019).

## 3.1 A-HPE Framework

The main component in the development of the near-optimal tensor methods of Gasnikov et al. (2019a); Bubeck et al. (2019); Jiang et al. (2019) is the Accelerated Hybrid Proximal Extragradient (A-HPE) framework of Monteiro and Svaiter (2013). This algorithmic framework can be seen as a generalization of the Accelerated Gradient Descent of Nesterov (1983). It is formalized as Algorithm 1. Next, we recall the main theorem by Monteiro and Svaiter (2013), which describes the convergence properties of Algorithm 1.

**Theorem 1** (Monteiro and Svaiter (2013)). *The iterations of Algorithm 1 satisfy the following inequality:*

$$2\beta_{K-1}(f(x_f^K) - f^*) + (1 - \sigma^2) \sum_{k=0}^{K-1} \alpha_k^{-2} \|x_f^{k+1} - x_g^k\|^2 \leq R^2. \tag{11}$$

Note that Algorithm 1 requires finding $x_f^{k+1}$ satisfying condition (8) on line 5. This condition can be rewritten as follows:

$$\|\nabla A_{\lambda_k}(x_f^{k+1}; x_g^k)\| \leq \sigma \lambda_k^{-1} \|x_f^{k+1} - x_g^k\|, \tag{12}$$

---

**Algorithm 2** Near-Optimal Tensor Method

---

1: **input:** $x^0 = x_f^0 \in \mathbb{R}^d$
2: **parameters:** $M > 0$, $K \in \{1, 2, \ldots\}$
3: $\beta_{-1} = 0$
4: **for** $k = 0, 1, 2, \ldots, K - 1$ **do**
5:     compute $\begin{cases} \lambda_k > 0 & \text{satisfying (17)} \\ x_f^{k+1} \in \mathbb{R}^d & \text{satisfying (15)} \\ x_g^k \in \mathbb{R}^d, \alpha_k \in (0, 1] & \text{satisfying (9)} \\ \eta_k, \beta_k > 0 & \text{satisfying (10)} \end{cases}$
6:     $x^{k+1} = x^k - \eta_k \nabla f(x_f^{k+1})$
7: **end for**
8: **output:** $x_f^K$

---

where function $A_\lambda(\cdot; z) \colon \mathbb{R}^d \to \mathbb{R}$ for $\lambda > 0$ and $z \in \mathbb{R}^d$ is defined as

$$A_\lambda(x; z) = f(x) + \frac{1}{2\lambda} \|x - z\|^2. \tag{13}$$

### 3.2 Application to High-Order Minimization

In order to perform the computation on line 5 of Algorithm 1, we need to find $x_f^{k+1} \in \mathbb{R}^d$ that satisfies condition (8). As we mentioned earlier, condition (8) is equivalent to (12), which involves the gradient norm $\|\nabla A_{\lambda_k}(\cdot; x_g^k)\|$ at point $x_f^{k+1}$. Function $A_{\lambda_k}(\cdot; x_g^k)$ has $L_p$-Lipschitz $p$-th order derivatives for $p \geq 2$ due to its definition (13) and Assumption 1.[7] Hence, it has the following upper bound, thanks to Theorem 1 of Nesterov (2021a):

$$A_{\lambda_k}(x; x_g^k) \leq \Phi_{A_{\lambda_k}(\cdot; x_g^k)}^p(x; x_g^k) + \frac{pM}{(p+1)!} \|x - x_g^k\|^{p+1}. \tag{14}$$

It turns out that $x_f^{k+1}$ can be obtained by minimizing this upper bound:

$$x_f^{k+1} = \underset{x \in \mathbb{R}^d}{\arg\min} \, \Phi_{A_{\lambda_k}(\cdot; x_g^k)}^p(x; x_g^k) + \frac{pM}{(p+1)!} \|x - x_g^k\|^{p+1}, \tag{15}$$

where $M > L_p$.[8] Indeed, by Lemma 1 of Nesterov (2021a), we have

$$\|\nabla A_{\lambda_k}(x_f^{k+1}; x_g^k)\| \leq \frac{pM + L_p}{p!} \|x_f^{k+1} - x_g^k\|^p. \tag{16}$$

Hence, to satisfy condition (12), we choose $\lambda_k$ in the following way:

$$\frac{\sigma p!}{2(pM + L_p)} \|x_f^{k+1} - x_g^k\|^{1-p} \leq \lambda_k \leq \frac{\sigma p!}{(pM + L_p)} \|x_f^{k+1} - x_g^k\|^{1-p}. \tag{17}$$

Here, the upper bound on $\lambda_k$ ensures condition (12), while the lower bound prevents stepsize $\lambda_k$ from being too small, which would hurt the convergence rate. The resulting near-optimal tensor method is formalized as Algorithm 2. It has the following convergence rate:

$$f(x_f^K) - f^* \leq \frac{\text{const} \cdot L_p \|x^0 - x^*\|^{p+1}}{K^{\frac{3p+1}{2}}}, \tag{18}$$

where $K$ is the number of iterations. The proof of this convergence rate involves condition (17) and Theorem 1. It is given in the works of Gasnikov et al. (2019a); Bubeck et al. (2019); Jiang et al. (2019).

---

[7]$\nabla^p A(x; x_g^k)[h] = \nabla^p f(x)[h]$ when $p > 2$, and $\nabla^2 A(x; x_g^k)[h] = \nabla^2 f(x)[h] + \lambda^{-1} \|h\|^2$.
[8]We require the strict inequality to ensure the uniform convexity of upper bound (14), which implies the uniqueness and the existence of the minimizer in (15).

### 3.3 The Problems with the Existing Algorithms

Algorithm 2 requires finding $\lambda_k$ satisfying condition (17) at each iteration. According to line 5 of Algorithm 2, $\lambda_k$ depends on $x_f^{k+1}$ via (17), which depends on $x_g^k$ via (15), which depends on $\eta_k, \beta_k$ via (9), which depend on $\lambda_k$ via (10). Hence, computation of stepsize $\lambda_k$ depends on $\lambda_k$ itself and there is no explicit way to perform the computation on line 5.

The algorithms of Gasnikov et al. (2019a); Bubeck et al. (2019); Jiang et al. (2019) use various binary search procedures to find $\lambda_k$ and perform the computation on line 5. However, such procedures are costly and require many iterations to converge. For instance, Bubeck et al. (2019) show that their variant of binary search requires the following number of $p$-th order oracle calls to find $\lambda_k$ satisfying condition (17):

$$\mathcal{O}\left(\log \frac{L_p R^{p+1}}{\epsilon}\right). \tag{19}$$

The same complexity (up to constant factors) for similar binary search procedures was established in the works of Nesterov (2021b); Jiang et al. (2019), and in the work of Monteiro and Svaiter (2013) for the $p = 2$ case. Hence, the total oracle complexity of Algorithm 2 is $\mathcal{O}\left(\epsilon^{-2/(3p+1)} \log(1/\epsilon)\right)$ which does not match the lower bound of Arjevani et al. (2019).

The additional logarithmic factor in the oracle complexity of Algorithm 2 raises the question whether it is superior to the accelerated tensor method of Nesterov (2021a) in practice. On the one hand, Gasnikov et al. (2019a) provided an experimental study that showed the practical superiority of Algorithm 2 over the algorithm of Nesterov (2021a). However, this experimental comparison is utterly unfair because it considers only the iteration complexity of the algorithms, which does not take into account the oracle complexity of the binary search procedure.

## 4 The First Optimal Tensor Method

In the previous section, we described the main issues with the existing high-order methods that prevent them from being optimal and practical algorithms for solving problem (1). In this section, we will show how to construct an algorithm that does not have those issues. More precisely, we will develop the first optimal $p$-th order algorithm ($p \geq 2$) for solving main problem (1).

### 4.1 The Key Idea

Gasnikov et al. (2019a); Bubeck et al. (2019); Jiang et al. (2019) used the following approach while creating their near-optimal algorithms: they fixed the procedure of computing $x_f^{k+1}$ on line 5 of Algorithm 1 using formula (15) and then developed the procedure for computing $\lambda_k$, which turned out to be inefficient. We will go the opposite way. That is, we choose parameters $\lambda_k$ in advance in such a way that they ensure the optimal convergence rate and then provide an efficient procedure for finding $x_f^{k+1}$ satisfying condition (8). Let $\eta_k$ be defined as follows:

$$\eta_k = \eta(1+k)^{\frac{3p-1}{2}}, \tag{20}$$

where $\eta > 0$ is a parameter. Using (10), we can compute $\beta_k$ and $\lambda_k$ as follows:

$$\beta_k = \eta \sum_{l=0}^{k}(1+l)^{\frac{3p-1}{2}}, \qquad \lambda_k = \frac{\eta(1+k)^{3p-1}}{\sum_{l=0}^{k}(1+l)^{\frac{3p-1}{2}}}. \tag{21}$$

The following lemma provides a lower bound on $\beta_k$ and an upper bound on $\lambda_k$.

**Lemma 1.** *Parameters $\beta_k$ and $\lambda_k$ defined by* (21) *satisfy the following inequalities:*

$$\beta_k \geq \frac{2\eta}{(3p+1)}(k+1)^{\frac{3p+1}{2}}, \qquad \lambda_k \leq \frac{\eta(3p+1)}{2}(1+k)^{\frac{3(p-1)}{2}}. \tag{22}$$

Lemma 1 and Theorem 1 immediately imply the convergence rate $\mathcal{O}(1/k^{(3p+1)/2})$, which matches the lower bound of Arjevani et al. (2019). Hence, the only remaining question is how to compute $x_f^{k+1}$ satisfying (8) efficiently. To be precise, we need to develop a procedure that can perform this computation using $\mathcal{O}(1)$ of $p$-th order oracle calls.

---

**Algorithm 3** Tensor Extragradient Method

---

1: **input:** $x^{k,0} = x_g^k \in \mathbb{R}^d$, $A^k(\cdot) = A_{\lambda_k}(\cdot; x_g^k)$
2: **parameters:** $M > 0$
3: $t = -1$
4: **repeat**
5:     $t = t + 1$
6:     compute $x^{k,t+1/2} \in \mathbb{R}^d$ as follows:

$$x^{k,t+1/2} = \underset{x \in \mathbb{R}^d}{\arg\min} \, \Phi_{A^k}^p(x; x^{k,t}) + \frac{pM}{(p+1)!} \|x - x^{k,t}\|^{p+1} \tag{23}$$

7:     $x^{k,t+1} = x^{k,t} - \left( \frac{M\|x^{k,t+1/2} - x^{k,t}\|^{p-1}}{(p-1)!} \right)^{-1} \nabla A^k(x^{k,t+1/2})$
8: **until** $\|\nabla A^k(x^{k,t+1/2})\| \leq \sigma \lambda_k^{-1} \|x^{k,t+1/2} - x^{k,0}\|$
9: $T^k = t + 1$
10: **output:** $x_f^{k+1} = x^{k,T^k-1/2}$

---

## 4.2 Tensor Extragradient Method for Gradient Norm Reduction

In this subsection, we develop an efficient procedure for computing $x_f^{k+1}$ satisfying condition (8). As we mentioned earlier, condition (8) is equivalent to (12), which is an upper bound on the gradient norm $\|\nabla A_{\lambda_k}(\cdot; x_g^k)\|$ at point $x_f^{k+1}$. Hence, we need an algorithm for the gradient norm reduction in the following smooth high-order convex minimization problem:

$$x^{k,*} = \underset{x \in \mathbb{R}^d}{\arg\min} \, A_{\lambda_k}(x; x_g^k). \tag{24}$$

In this subsection, we provide such an algorithm. We call the algorithm Tensor Extragradient Method. It is formalized as Algorithm 3. In the case $p = 1$, this algorithm recovers the extragradient method of Korpelevich (1976). Algorithm 3 can be seen as a generalization of the extragradient method for high-order optimization.

One can observe that due to line 8 of Algorithm 3, $x_f^{k+1} = x^{k,T^k-1/2}$ satisfies condition (12), where $x^{k,T^k-1/2}$ is the output of Algorithm 3. This is exactly what we need. The following theorem provides an upper bound on the number of iterations $T^k$ required by Algorithm 3 to terminate and produce the output $x_f^{k+1}$.

**Theorem 2.** *Let $M$ satisfy*

$$M \geq L_p. \tag{25}$$

*Then step (23) on line 6 of Algorithm 3 is well defined and the number of iterations $T^k$ performed by Algorithm 3 is upper-bounded as follows:*

$$T^k \leq \left( \lambda_k C_p(M, \sigma) \|x_g^k - x^{k,*}\|^{p-1} \right)^{2/p} + 1, \tag{26}$$

*where $C_p$ is defined as*

$$C_p(M, \sigma) = \frac{p^p M^p (1 + \sigma^{-1})}{p!(pM - L_p)^{p/2}(pM + L_p)^{p/2-1}}. \tag{27}$$

Algorithm 3 and Theorem 2 will further be used for the construction of the optimal high-order algorithm for solving problem (1). It is worth mentioning the potential alternatives to Algorithm 3 that we could use for gradient norm reduction. For instance, we could use the tensor method of Nesterov (2021a). However, the upper bound on the number of iterations for this method would involve the diameter of the level set of function $A_{\lambda_k}(\cdot; x_g^k)$ rather than the distance to the solution $\|x_g^k - x^{k,*}\|$. This would be an obstacle towards development of the optimal algorithm. Alternatively, we could use the accelerated tensor method of Nesterov (2021a). It turns out that it would work as we need. Moreover, the upper bound on the number of iterations would be even better than (26). However, we find the accelerated tensor method of Nesterov (2021a) to be too complicated, which could make the resulting optimal high-order method hard to implement. On the other hand, it would not give us any benefits for the construction of the optimal high-order method compared to Algorithm 3.

---

**Algorithm 4** Optimal Tensor Method

---
1: **input:** $x^0 = x_f^0 \in \mathbb{R}^d$
2: **parameters:** $\eta > 0, M > 0, \sigma \in (0,1), K \in \{1, 2, \ldots\}$
3: $\beta_{-1} = 0$
4: **for** $k = 0, 1, 2, \ldots, K - 1$ **do**
5: $\quad \eta_k = \eta(1 + k)^{(3p-1)/2}$
6: $\quad \beta_k = \beta_{k-1} + \eta_k, \lambda_k = \eta_k^2/\beta_k, \alpha_k = \eta_k/\beta_k$
7: $\quad x_g^k = \alpha_k x^k + (1 - \alpha_k)x_f^k$
8: $\quad x^{k,0} = x_g^k, t = -1$
9: $\quad$ **repeat**
10: $\qquad t = t + 1$
11: $\qquad x^{k,t+1/2} = \arg\min_{x \in \mathbb{R}^d} \Phi_{A_{\lambda_k}(\cdot; x_g^k)}^p(x; x^{k,t}) + \frac{pM}{(p+1)!}\|x - x^{k,t}\|^{p+1}$
12: $\qquad x^{k,t+1} = x^{k,t} - \left(\frac{M\|x^{k,t+1/2}-x^{k,t}\|^{p-1}}{(p-1)!}\right)^{-1}\nabla A_{\lambda_k}(x^{k,t+1/2}; x_g^k)$
13: $\quad$ **until** $\|\nabla A_{\lambda_k}(x^{k,t+1/2}; x_g^k)\| \leq \sigma\lambda_k^{-1}\|x^{k,t+1/2} - x^{k,0}\|$
14: $\quad T^k = t + 1$
15: $\quad x_f^{k+1} = x^{k,T^k-1/2}$
16: $\quad x^{k+1} = x^k - \eta_k\nabla f(x_f^{k+1})$
17: **end for**
18: **output:** $x_f^K$

---

## 4.3 Modification of the Analysis of A-HPE Framework

Unfortunately, we cannot use Theorem 1 for the analysis of our optimal algorithm. This is because inequality (11) involves the distances $\|x_g^k - x_f^{k+1}\|$ on the right-hand side. Hence, inequality (11) does not allow us to estimate the iteration complexity $T^k$ of Algorithm 3 using Theorem 2. Further, we provide a new theorem that includes the analysis of the A-HPE framework and provides an upper bound on the distances $\|x_g^k - x^{k,*}\|$.

**Theorem 3.** *The iterations of Algorithm 1 satisfy the following inequality:*

$$2\beta_{K-1}(f(x_f^K) - f^*) + \frac{1-\sigma}{1+\sigma}\sum_{k=0}^{K-1}\alpha_k^{-2}\|x_g^k - x^{k,*}\|^2 \leq R^2. \tag{28}$$

## 4.4 The First Optimal Tensor Method

Now, we are ready to provide the first optimal high-order algorithm for solving problem (1). In order to construct this algorithm, we use our Tensor Extragradient Method (Algorithm 3) to perform the computations on line 5 of the A-HPE Framework (Algorithm 1). We also use our choice of parameters $\eta_k, \beta_k$ and $\lambda_k$ which is provided by (20) and (21). The resulting algorithm is formalized as Algorithm 4.

Now, we are ready to prove that Algorithm 4 is an optimal algorithm. First, we need to establish an upper bound on the number of iterations $T^k$ performed by the inner repeat-loop of Algorithm 4. This is done by the following theorem.

**Theorem 4.** *Let $M$ satisfy* (25). *Then, the following inequality holds for Algorithm 4:*

$$\sum_{k=0}^{K-1} T^k \leq K + (1 + K)\left(\frac{\eta(3p+1)^p C_p(M,\sigma)R^{p-1}}{2^p\sqrt{p}} \cdot \left(\frac{1+\sigma}{1-\sigma}\right)^{\frac{p-1}{2}}\right)^{\frac{2}{p}}, \tag{29}$$

*where $C_p$ is defined by* (27).

Theorem 4 implies that with a proper choice of the parameter $\eta$, Algorithm 4 performs $\mathcal{O}(1)$ $p$-th order oracle calls per iteration on average. Indeed, let $\eta$ be chosen as follows:

$$\eta = \left(\frac{(3p+1)^p C_p(M,\sigma)R^{p-1}}{2^p\sqrt{p}} \cdot \left(\frac{1+\sigma}{1-\sigma}\right)^{\frac{p-1}{2}}\right)^{-1}. \tag{30}$$

Then, Theorem 4 immediately implies

$$\sum_{k=0}^{K-1} T^k \leq 2K + 1. \tag{31}$$

Finally, the following theorem establishes the total $p$-th order oracle complexity of Algorithm 4.

**Theorem 5.** *Let $M = L_p$ and $\sigma = 1/2$. Let $\eta$ be defined by (30). Then, to reach precision $f(x_f^k) - f^* \leq \epsilon$, Algorithm 4 requires no more than the following number of $p$-th order oracle calls:*

$$5D_p \cdot \left(L_p R^{p+1}/\epsilon\right)^{\frac{2}{3p+1}} + 7, \tag{32}$$

*where $D_p$ is defined as follows:*

$$D_p = \left(\frac{3^{\frac{p+1}{2}}(3p+1)^{p+1}p^p(p+1)}{2^{p+2}\sqrt{p}p!(p^2-1)^{\frac{p}{2}}}\right)^{\frac{2}{3p+1}}. \tag{33}$$

Theorem 5 shows that the total $p$-th order oracle complexity of Algorithm 4 is $\mathcal{O}\left(\left(L_p R^{p+1}/\epsilon\right)^{\frac{2}{3p+1}}\right)$. This oracle complexity matches the lower bounds of Arjevani et al. (2019) up to a universal constant that does not depend on $R$, $L_p$ and $\epsilon$. Hence, Algorithm 4 is indeed the first optimal high-order algorithm for solving smooth convex minimization problems.

### 4.5 Practical Performance

In this paper we provide an experimental comparison of the proposed optimal high-order algorithm for solving smooth convex minimization problems (Algorithm 4) with the existing near-optimal high-order method of Gasnikov et al. (2019a); Bubeck et al. (2019); Jiang et al. (2019) (Algorithm 2). In summary, the experiments show that the proposed optimal Algorithm 4 is indeed a practical algorithm which significantly outperforms the existing near-optimal Algorithm 2. The details can be found in Appendix F.

## Acknowledgments

The work of Alexander Gasnikov was supported by the strategic academic leadership program 'Priority 2030' (Agreement 075-02-2021-1316 30.09.2021).

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
