# Appendix

## A   Proof of Lemma 1

The lower bound on $\beta_k$ can be obtained in the following way

$$\beta_k = \sum_{l=0}^{k} \eta(1+l)^{\frac{3p-1}{2}} = \sum_{l=0}^{k} \eta \int_0^1 (1+l)^{\frac{3p-1}{2}} dt \geq \sum_{l=0}^{k} \eta \int_0^1 (t+l)^{\frac{3p-1}{2}} dt$$

$$= \sum_{l=0}^{k} \eta \int_l^{l+1} t^{\frac{3p-1}{2}} dt = \eta \int_0^{k+1} t^{\frac{3p-1}{2}} dt = \frac{2\eta}{(3p+1)} (k+1)^{\frac{3p+1}{2}}.$$

Upper bound on $\lambda_k$ is obtained using the lower bound on $\beta_k$ and (10). $\qquad\square$

## B   Proof of Theorem 2

$$\|x^{k,t+1} - x^{k,*}\|^2 = \|x^{k,t} - x^{k,*}\|^2 + 2\langle x^{k,t+1} - x^{k,t}, x^{k,t} - x^{k,*}\rangle + \|x^{k,t+1} - x^{k,t}\|^2$$

$$= \|x^{k,t} - x^{k,*}\|^2 + 2\langle x^{k,t+1} - x^{k,t}, x^{k,t+1/2} - x^{k,*}\rangle$$

$$+ 2\langle x^{k,t+1} - x^{k,t}, x^{k,t} - x^{k,t+1/2}\rangle + \|x^{k,t+1} - x^{k,t}\|^2$$

$$= \|x^{k,t} - x^{k,*}\|^2 + 2\langle x^{k,t+1} - x^{k,t}, x^{k,t+1/2} - x^{k,*}\rangle$$

$$+ \|x^{k,t+1} - x^{k,t+1/2}\|^2 - \|x^{k,t} - x^{k,t+1/2}\|^2.$$

From (23) online 6 of Algorithm 3, we have

$$x^{k,t+1/2} = x^{k,t} - \left( \frac{M\|x^{k,t+1/2} - x^{k,t}\|^{p-1}}{(p-1)!} \right)^{-1} \nabla\Phi_{A^k}^p(x^{k,t+1/2}; x^{k,t}).$$

Plugging this into the previous equation and using line 7 of Algorithm 3, we get

$$\|x^{k,t+1} - x^{k,*}\|^2 = \|x^{k,t} - x^{k,*}\|^2 - 2\gamma_{k,t}\langle \nabla A^k(x^{k,t+1/2}), x^{k,t+1/2} - x^{k,*}\rangle$$

$$+ \gamma_{k,t}^2 \|\nabla\Phi_{A^k}^p(x^{k,t+1/2}; x^{k,t}) - \nabla A^k(x^{k,t+1/2})\|^2 - \|x^{k,t} - x^{k,t+1/2}\|^2,$$

where $\gamma_{k,t} = \left( \frac{M\|x^{k,t+1/2}-x^{k,t}\|^{p-1}}{(p-1)!} \right)^{-1}$. Using the convexity of function $A^k(x)$, we get

$$\|x^{k,t+1} - x^{k,*}\|^2 = \|x^{k,t} - x^{k,*}\|^2 - 2\gamma_{k,t}(A^k(x^{k,t+1/2}) - A^k(x^{k,*}))$$

$$+ \gamma_{k,t}^2 \|\nabla\Phi_{A^k}^p(x^{k,t+1/2}; x^{k,t}) - \nabla A^k(x^{k,t+1/2})\|^2 - \|x^{k,t} - x^{k,t+1/2}\|^2$$

$$\leq \|x^{k,t} - x^{k,*}\|^2 + \gamma_{k,t}^2 \|\nabla\Phi_{A^k}^p(x^{k,t+1/2}; x^{k,t}) - \nabla A^k(x^{k,t+1/2})\|^2$$

$$- \|x^{k,t} - x^{k,t+1/2}\|^2.$$

Using inequality (1.6) of Nesterov (2021a), we get

$$\|x^{k,t+1} - x^{k,*}\|^2 \leq \|x^{k,t} - x^{k,*}\|^2 + \left( \frac{\gamma_{k,t}L_p}{p!} \right)^2 \|x^{k,t+1/2} - x^{k,t}\|^{2p} - \|x^{k,t} - x^{k,t+1/2}\|^2$$

$$= \|x^{k,t} - x^{k,*}\|^2 - \left( 1 - \left( \frac{\gamma_{k,t}L_p}{p!}\|x^{k,t+1/2} - x^{k,t}\|^{p-1} \right)^2 \right) \|x^{k,t} - x^{k,t+1/2}\|^2$$

$$= \|x^{k,t} - x^{k,*}\|^2 - \left( 1 - \left( \frac{L_p(p-1)!}{p!M} \right)^2 \right) \|x^{k,t} - x^{k,t+1/2}\|^2$$

$$= \|x^{k,t} - x^{k,*}\|^2 - \left( 1 - \left( \frac{L_p}{pM} \right)^2 \right) \|x^{k,t} - x^{k,t+1/2}\|^2.$$

Using Lemma 1 of Nesterov (2021a), we get

$$\|x^{k,t+1} - x^{k,*}\|^2 \le \|x^{k,t} - x^{k,*}\|^2 - \left(1 - \left(\frac{L_p}{pM}\right)^2\right)\left(\frac{p!}{pM + L_p}\|\nabla A^k(x^{k,t+1/2})\|\right)^{2/p}.$$

After telescoping and rearranging, for $T \le T^k$ we get

$$T \min_{t \in \{0,1,\dots,T-1\}} \frac{(pM - L_p)(pM + L_p)}{p^2 M^2}\left(\frac{p!}{pM + L_p}\|\nabla A^k(x^{k,t+1/2})\|\right)^{2/p} \le \|x^{k,0} - x^{k,*}\|^2.$$

Taking both sides of the inequality in the power of $p/2$ gives

$$\|x^{k,0} - x^{k,*}\|^p \ge T^{p/2} \min_{t \in \{0,1,\dots,T-1\}} \frac{(pM - L_p)^{p/2}(pM + L_p)^{p/2}}{p^p M^p} \frac{p!}{pM + L_p}\|\nabla A^k(x^{k,t+1/2})\|$$

$$= T^{p/2} \min_{t \in \{0,1,\dots,T-1\}} \frac{p!(pM - L_p)^{p/2}(pM + L_p)^{p/2-1}}{p^p M^p}\|\nabla A^k(x^{k,t+1/2})\|.$$

After rearranging, we get

$$\min_{t \in \{0,1,\dots,T-1\}} \|\nabla A^k(x^{k,t+1/2})\| \le \frac{p^p M^p \|x^{k,0} - x^{k,*}\|^p}{p!(pM - L_p)^{p/2}(pM + L_p)^{p/2-1}} \cdot \frac{1}{T^{p/2}}.$$

Now, let us prove upper bound (26) by a contradiction. Suppose that (26) is not true. Hence,

$$T^k > \left(\frac{\lambda_k p^p M^p (1 + \sigma^{-1})\|x^{k,0} - x^{k,*}\|^{p-1}}{p!(pM - L_p)^{p/2}(pM + L_p)^{p/2-1}}\right)^{2/p} + 1.$$

This implies

$$\min_{t \in \{0,1,\dots,T-1\}} \left(\|\nabla A^k(x^{k,t+1/2})\| - c\lambda_k^{-1}\|x^{k,0} - x^{k,*}\|\right) \le 0,$$

where $c = (1 + \sigma^{-1})^{-1}$ and $T = T^k - 1$. Using the $\lambda_k^{-1}$-strong convexity of $A^k(x)$, we get

$$0 \ge \min_{t \in \{0,1,\dots,T-1\}} \left(\|\nabla A^k(x^{k,t+1/2})\| - c\lambda_k^{-1}\|x^{k,0} - x^{k,*}\|\right)$$

$$\ge \min_{t \in \{0,1,\dots,T-1\}} \left(\|\nabla A^k(x^{k,t+1/2})\| - c\lambda_k^{-1}\|x^{k,0} - x^{k,t+1/2}\| - c\lambda_k^{-1}\|x^{k,*} - x^{k,t+1/2}\|\right)$$

$$\ge \min_{t \in \{0,1,\dots,T-1\}} \left(\|\nabla A^k(x^{k,t+1/2})\| - c\lambda_k^{-1}\|x^{k,0} - x^{k,t+1/2}\| - c\|\nabla A^k(x^{k,t+1/2})\|\right)$$

$$= \min_{t \in \{0,1,\dots,T-1\}} \left((1 - c)\|\nabla A^k(x^{k,t+1/2})\| - c\lambda_k^{-1}\|x^{k,0} - x^{k,t+1/2}\|\right).$$

Dividing by $1 - c$ gives

$$0 \ge \min_{t \in \{0,1,\dots,T-1\}} \left(\|\nabla A^k(x^{k,t+1/2})\| - \frac{\lambda_k^{-1}}{c^{-1} - 1}\|x^{k,0} - x^{k,t+1/2}\|\right).$$

Plugging $c = (1 + \sigma^{-1})^{-1}$ gives

$$0 \ge \min_{t \in \{0,1,\dots,T-1\}} \left(\|\nabla A^k(x^{k,t+1/2})\| - \sigma\lambda_k^{-1}\|x^{k,0} - x^{k,t+1/2}\|\right).$$

This means that the inner repeat-loop of Algorithm 3 terminated after no more than $T$ iterations, which contradicts with $T^k = T + 1$. This concludes the proof. □

## C   Proof of Theorem 3

Here we also provide the proof of Theorem 1 for completeness. Using line 6 of Algorithm 1, we get

$$\|x^{k+1} - x^*\|^2 = \|x^k - x^*\|^2 - 2\eta_k\langle\nabla f(x_f^{k+1}), x^k - x^*\rangle + \eta_k^2\|\nabla f(x_f^{k+1})\|^2.$$

Using (9), we get $x^k = \alpha_k^{-1} x_g^k - \alpha_k^{-1}(1 - \alpha_k) x_f^k$, which implies

$$\|x^{k+1} - x^*\|^2 = \|x^k - x^*\|^2 - 2\eta_k \langle \nabla f(x_f^{k+1}), \alpha_k^{-1} x_g^k - \alpha_k^{-1}(1 - \alpha_k) x_f^k - x^* \rangle + \eta_k^2 \|\nabla f(x_f^{k+1})\|^2$$

$$= \|x^k - x^*\|^2 + \eta_k^2 \|\nabla f(x_f^{k+1})\|^2$$

$$+ 2(\beta_k - \eta_k)\langle \nabla f(x_f^{k+1}), x_f^k \rangle - 2\beta_k \langle \nabla f(x_f^{k+1}), x_g^k \rangle + 2\eta_k \langle \nabla f(x_f^{k+1}), x^* \rangle$$

$$= \|x^k - x^*\|^2 + 2(\beta_k - \eta_k)\langle \nabla f(x_f^{k+1}), x_f^k - x_f^{k+1} \rangle + 2\eta_k \langle \nabla f(x_f^{k+1}), x^* - x_f^{k+1} \rangle$$

$$- 2\beta_k \langle \nabla f(x_f^{k+1}), x_g^k - x_f^{k+1} \rangle + \eta_k^2 \|\nabla f(x_f^{k+1})\|^2.$$

Using the convexity of $f(x)$ and (10), we get

$$\|x^{k+1} - x^*\|^2 \leq \|x^k - x^*\|^2 + 2(\beta_k - \eta_k)(f(x_f^k) - f(x_f^{k+1})) + 2\eta_k(f^* - f(x_f^{k+1}))$$

$$- 2\beta_k \langle \nabla f(x_f^{k+1}), x_g^k - x_f^{k+1} \rangle + \eta_k^2 \|\nabla f(x_f^{k+1})\|^2$$

$$= \|x^k - x^*\|^2 - \beta_k(f(x_f^{k+1}) - f^*) + \beta_{k-1}(f(x_f^k) - f^*)$$

$$- 2\beta_k \langle \nabla f(x_f^{k+1}), x_g^k - x_f^{k+1} \rangle + \eta_k^2 \|\nabla f(x_f^{k+1})\|^2.$$

Using (9), we get

$$\|x^{k+1} - x^*\|^2 \leq \|x^k - x^*\|^2 - \beta_k(f(x_f^{k+1}) - f^*) + \beta_{k-1}(f(x_f^k) - f^*)$$

$$- 2\langle \eta_k \nabla f(x_f^{k+1}), \beta_k \eta_k^{-1}(x_g^k - x_f^{k+1}) \rangle + \eta_k^2 \|\nabla f(x_f^{k+1})\|^2$$

$$= \|x^k - x^*\|^2 - \beta_k(f(x_f^{k+1}) - f^*) + \beta_{k-1}(f(x_f^k) - f^*)$$

$$+ 2\langle \eta_k \nabla f(x_f^{k+1}), \alpha_k^{-1}(x_f^{k+1} - x_g^k) \rangle + \eta_k^2 \|\nabla f(x_f^{k+1})\|^2.$$

Using the parallelogram rule, we get

$$\|x^{k+1} - x^*\|^2 \leq \|x^k - x^*\|^2 - \beta_k(f(x_f^{k+1}) - f^*) + \beta_{k-1}(f(x_f^k) - f^*)$$

$$+ \|\eta_k \nabla f(x_f^{k+1}) + \alpha_k^{-1}(x_f^{k+1} - x_g^k)\|^2 - \alpha_k^{-2}\|x_f^{k+1} - x_g^k\|^2$$

$$= \|x^k - x^*\|^2 - \beta_k(f(x_f^{k+1}) - f^*) + \beta_{k-1}(f(x_f^k) - f^*)$$

$$+ \eta_k^2 \|\nabla f(x_f^{k+1}) + \eta_k^{-1}\alpha_k^{-1}(x_f^{k+1} - x_g^k)\|^2 - \alpha_k^{-2}\|x_f^{k+1} - x_g^k\|^2.$$

Using (9) and (10), we get

$$\|x^{k+1} - x^*\|^2 \leq \|x^k - x^*\|^2 - \beta_k(f(x_f^{k+1}) - f^*) + \beta_{k-1}(f(x_f^k) - f^*)$$

$$+ \eta_k^2 \|\nabla f(x_f^{k+1}) + \lambda_k^{-1}(x_f^{k+1} - x_g^k)\|^2 - \alpha_k^{-2}\|x_f^{k+1} - x_g^k\|^2.$$

Using (8), we get

$$\|x^{k+1} - x^*\|^2 \leq \|x^k - x^*\|^2 - \beta_k(f(x_f^{k+1}) - f^*) + \beta_{k-1}(f(x_f^k) - f^*)$$

$$+ \eta_k^2 \lambda_k^{-2} \sigma^2 \|x_f^{k+1} - x_g^k\|^2 - \alpha_k^{-2}\|x_f^{k+1} - x_g^k\|^2$$

$$= \|x^k - x^*\|^2 - \beta_k(f(x_f^{k+1}) - f^*) + \beta_{k-1}(f(x_f^k) - f^*)$$

$$- \alpha_k^{-2}(1 - \sigma^2)\|x_f^{k+1} - x_g^k\|^2.$$

Now, let us bound $\|x_g^k - x^{k,*}\|$ using $\lambda_k^{-1}$-strong convexity of $A_{\lambda_k}(\cdot; x_g^k)$ and (8):

$$\|x_g^k - x^{k,*}\| \leq \|x_g^k - x_f^{k+1}\| + \|x_f^{k+1} - x^{k,*}\|$$

$$\leq \|x_g^k - x_f^{k+1}\| + \lambda_k \|\nabla A_{\lambda_k}(\cdot; x_g^k)\|$$

$$\leq (1 + \sigma)\|x_f^{k+1} - x_g^k\|.$$

Plugging this into the previous inequality gives

$$\|x^{k+1} - x^*\|^2 \leq \|x^k - x^*\|^2 - \beta_k(f(x_f^{k+1}) - f^*) + \beta_{k-1}(f(x_f^k) - f^*)$$

$$- \alpha_k^{-2}\frac{(1 - \sigma^2)}{(1 + \sigma)^2}\|x_g^k - x^{k,*}\|^2$$

$$= \|x^k - x^*\|^2 - \beta_k(f(x_f^{k+1}) - f^*) + \beta_{k-1}(f(x_f^k) - f^*)$$

$$- \alpha_k^{-2}\frac{(1 - \sigma)}{(1 + \sigma)}\|x_g^k - x^{k,*}\|^2.$$

Rearranging and telescoping concludes the proof. $\qquad\square$

## D  Proof of Theorem 4

Using Theorem 2, we get

$$\left(\sum_{k=0}^{K-1}(T^k-1)\right)^{\frac{p}{p-1}} \le \left(\sum_{k=0}^{K-1}\left(\lambda_k C_p(M,\sigma)\|x_g^k - x^{k,*}\|^{p-1}\right)^{2/p}\right)^{\frac{p}{p-1}}.$$

Let us choose parameters $\tau_0,\ldots,\tau_{K-1}$ as follows:

$$\tau_k = \left(\sum_{l=0}^{K-1}(1+l)^{p-1}\right)^{-1}(1+k)^{p-1}$$

Then, we have

$$\left(\sum_{k=0}^{K-1}(T^k-1)\right)^{\frac{p}{p-1}} \le \left(\sum_{k=0}^{K-1}\tau_k\tau_k^{-1}\left(\lambda_k C_p(M,\sigma)\|x_g^k - x^{k,*}\|^{p-1}\right)^{2/p}\right)^{\frac{p}{p-1}}.$$

Note that parameters $\tau_k$ satisfy

$$\sum_{k=0}^{K-1}\tau_k = 0, \qquad \tau_0,\ldots,\tau_{K-1} \ge 0.$$

Hence, using the convexity of function $(\cdot)^{p/(p-1)}$, we get

$$\left(\sum_{k=0}^{K-1}(T^k-1)\right)^{\frac{p}{p-1}} \le \sum_{k=0}^{K-1}\tau_k\left(\tau_k^{-1}\left(\lambda_k C_p(M,\sigma)\|x_g^k - x^{k,*}\|^{p-1}\right)^{2/p}\right)^{\frac{p}{p-1}}$$

$$= C_p(M,\sigma)^{\frac{2}{p-1}}\sum_{k=0}^{K-1}\tau_k^{\frac{-1}{p-1}}\left(\lambda_k\right)^{\frac{2}{p-1}}\|x_g^k - x^{k,*}\|^2.$$

Using Lemma 1, we get

$$\left(\sum_{k=0}^{K-1}(T^k-1)\right)^{\frac{p}{p-1}} \le C_p(M,\sigma)^{\frac{2}{p-1}}\sum_{k=0}^{K-1}\tau_k^{\frac{-1}{p-1}}\left(\frac{\eta(3p+1)}{2}(1+k)^{\frac{3(p-1)}{2}}\right)^{\frac{2}{p-1}}\|x_g^k - x^{k,*}\|^2$$

$$= \left(\frac{\eta(3p+1)C_p(M,\sigma)}{2}\right)^{\frac{2}{p-1}}\sum_{k=0}^{K-1}\tau_k^{\frac{-1}{p-1}}(1+k)^3\|x_g^k - x^{k,*}\|^2.$$

Using the definition of $\tau_k$, we get

$$\left(\sum_{k=0}^{K-1}(T^k-1)\right)^{\frac{p}{p-1}} \le \left(\frac{\eta(3p+1)C_p(M,\sigma)}{2}\right)^{\frac{2}{p-1}}\cdot\left(\sum_{l=0}^{K-1}(1+l)^{p-1}\right)^{\frac{1}{p-1}}$$

$$\cdot\sum_{k=0}^{K-1}(1+k)^2\|x_g^k - x^{k,*}\|^2.$$

Using the inequality

$$\sum_{l=0}^{K-1}(1+l)^{p-1} \le \sum_{l=0}^{K-1}\int_0^1(1+l+t)^{p-1}dt = \sum_{l=0}^{K-1}\int_{l+1}^{l+2}t^{p-1}dt$$

$$= \int_1^{K+1}t^{p-1}dt = \frac{(1+K)^p - 1}{p} \le \frac{1}{p}(1+K)^p,$$

we get

$$\left(\sum_{k=0}^{K-1}(T^k-1)\right)^{\frac{p}{p-1}} \le \left(\frac{\eta(3p+1)C_p(M,\sigma)}{2}\right)^{\frac{2}{p-1}} \cdot \left(\frac{1}{p}(1+K)^p\right)^{\frac{1}{p-1}}$$

$$\cdot \sum_{k=0}^{K-1}(1+k)^2\|x_g^k-x^{k,*}\|^2$$

$$= \left(\frac{\eta(3p+1)C_p(M,\sigma)}{2\sqrt{p}}\right)^{\frac{2}{p-1}}(1+K)^{\frac{p}{p-1}}\sum_{k=0}^{K-1}(1+k)^2\|x_g^k-x^{k,*}\|^2.$$

From (9) and Lemma 1, we get

$$\alpha_k^{-1} = \frac{\beta_k}{\eta_k} \ge \frac{2\eta(1+k)^{\frac{3p+1}{2}}}{(3p+1)}\cdot\frac{1}{\eta(1+k)^{\frac{3p-1}{2}}} = \frac{2}{(3p+1)}(1+k).$$

Hence,

$$\left(\sum_{k=0}^{K-1}(T^k-1)\right)^{\frac{p}{p-1}} \le \left(\frac{\eta(3p+1)C_p(M,\sigma)}{2\sqrt{p}}\right)^{\frac{2}{p-1}}(1+K)^{\frac{p}{p-1}}\sum_{k=0}^{K-1}\frac{(3p+1)^2}{4}\alpha_k^{-2}\|x_g^k-x^{k,*}\|^2$$

$$= \left(\frac{\eta(3p+1)C_p(M,\sigma)}{2\sqrt{p}}\right)^{\frac{2}{p-1}}\frac{(3p+1)^2(1+K)^{\frac{p}{p-1}}}{4}\sum_{k=0}^{K-1}\alpha_k^{-2}\|x_g^k-x^{k,*}\|^2.$$

Using Theorem 3, we get

$$\left(\sum_{k=0}^{K-1}(T^k-1)\right)^{\frac{p}{p-1}} \le \left(\frac{\eta(3p+1)C_p(M,\sigma)}{2\sqrt{p}}\right)^{\frac{2}{p-1}}\frac{(3p+1)^2(1+K)^{\frac{p}{p-1}}}{4}\frac{1+\sigma}{1-\sigma}R^2.$$

After taking both sides of the inequality in the power of $\frac{p-1}{p}$ we get

$$\sum_{k=0}^{K-1}(T^k-1) \le \left(\frac{\eta(3p+1)C_p(M,\sigma)}{2\sqrt{p}}\right)^{\frac{2}{p}}\frac{(3p+1)^{\frac{2(p-1)}{p}}(1+K)}{2^{\frac{2(p-1)}{p}}}\left(\frac{1+\sigma}{1-\sigma}R^2\right)^{\frac{p-1}{p}}$$

$$= (1+K)\left(\frac{\eta(3p+1)^pC_p(M,\sigma)R^{p-1}}{2^p\sqrt{p}}\cdot\left(\frac{1+\sigma}{1-\sigma}\right)^{\frac{p-1}{2}}\right)^{\frac{2}{p}}.$$

After rearranging, we get

$$\sum_{k=0}^{K-1}T^k \le K + (1+K)\left(\frac{\eta(3p+1)^pC_p(M,\sigma)R^{p-1}}{2^p\sqrt{p}}\cdot\left(\frac{1+\sigma}{1-\sigma}\right)^{\frac{p-1}{2}}\right)^{\frac{2}{p}}.$$

$\square$

## E   Proof of Theorem 5

Theorem 3 implies

$$f(x_f^K)-f^* \le R^2/(2\beta_{K-1}).$$

Using Lemma 1, we get

$$f(x_f^K)-f^* \le \frac{(3p+1)R^2}{4\eta}\cdot\frac{1}{K^{\frac{3p+1}{2}}}.$$

Choosing $K = \left\lceil \left( \frac{(3p+1)R^2}{4\eta\epsilon} \right)^{\frac{2}{3p+1}} \right\rceil$ implies $f(x_f^K) - f^* \le \epsilon$. Hence, we have the following upper bound on the total iteration complexity of Algorithm 4:

$$K \le \left\lceil \left( \frac{(3p+1)R^2}{4\eta\epsilon} \right)^{\frac{2}{3p+1}} \right\rceil$$

$$\le \left( \frac{(3p+1)R^2}{4\eta\epsilon} \right)^{\frac{2}{3p+1}} + 1.$$

Plugging $\eta$ defined by (30) gives

$$K \le \left( \frac{(3p+1)R^2}{4\epsilon} \cdot \frac{(3p+1)^p C_p(M,\sigma)R^{p-1}}{2^p\sqrt{p}} \cdot \left( \frac{1+\sigma}{1-\sigma} \right)^{\frac{p-1}{2}} \right)^{\frac{2}{3p+1}} + 1.$$

$$= \left( \frac{(3p+1)^{p+1}C_p(M,\sigma)R^{p+1}}{2^{p+2}\sqrt{p}\epsilon} \cdot \left( \frac{1+\sigma}{1-\sigma} \right)^{\frac{p-1}{2}} \right)^{\frac{2}{3p+1}} + 1.$$

Using the definition of $C_p(M,\sigma)$, we get

$$K \le \left( \frac{(3p+1)^{p+1}R^{p+1}}{2^{p+2}\sqrt{p}\epsilon} \cdot \left( \frac{1+\sigma}{1-\sigma} \right)^{\frac{p-1}{2}} \cdot \frac{p^p M^p(1+\sigma^{-1})}{p!(pM-L_p)^{p/2}(pM+L_p)^{p/2-1}} \right)^{\frac{2}{3p+1}}$$
$$+ 1.$$

Using $M = L_p$, we get

$$K \le \left( \frac{(3p+1)^{p+1}R^{p+1}}{2^{p+2}\sqrt{p}\epsilon} \cdot \left( \frac{1+\sigma}{1-\sigma} \right)^{\frac{p-1}{2}} \cdot \frac{p^p L_p(1+\sigma^{-1})}{p!(p-1)^{p/2}(p+1)^{p/2-1}} \right)^{\frac{2}{3p+1}} + 1$$

$$= \left( \frac{L_p R^{p+1}}{\epsilon} \cdot \frac{(3p+1)^{p+1}p^p(p+1)}{2^{p+2}\sqrt{p}p!(p^2-1)^{\frac{p}{2}}} \cdot \frac{(1+\sigma)^{\frac{p+1}{2}}}{\sigma(1-\sigma)^{\frac{p-1}{2}}} \right)^{\frac{2}{3p+1}} + 1$$

$$= \left( \frac{L_p R^{p+1}}{\epsilon} \right)^{\frac{2}{3p+1}} \cdot \left( \frac{(3p+1)^{p+1}p^p(p+1)}{2^{p+2}\sqrt{p}p!(p^2-1)^{\frac{p}{2}}} \cdot \frac{(1+\sigma)^{\frac{p+1}{2}}}{\sigma(1-\sigma)^{\frac{p-1}{2}}} \right)^{\frac{2}{3p+1}} + 1.$$

Using $\sigma = 1/2$, we get

$$K \le \left( \frac{L_p R^{p+1}}{\epsilon} \right)^{\frac{2}{3p+1}} \cdot \left( \frac{(3p+1)^{p+1}p^p(p+1)}{2^{p+2}\sqrt{p}p!(p^2-1)^{\frac{p}{2}}} \cdot 3^{\frac{p+1}{2}} \right)^{\frac{2}{3p+1}} + 1$$

$$= D_p \cdot \left( \frac{L_p R^{p+1}}{\epsilon} \right)^{\frac{2}{3p+1}} + 1.$$

Finally, we have that Algorithm 4 performs $(1 + 2T^k)$ of $p$-th order oracle calls at each iteration. Hence, using (31), we get following upper bound on the total oracle complexity:

$$\sum_{k=0}^{K-1} (1 + 2T^k) \le K + 2(2K+1) = 5K + 2$$

$$\le 5D_p \cdot \left( \frac{L_p R^{p+1}}{\epsilon} \right)^{\frac{2}{3p+1}} + 7.$$

$\square$

# F  Experiments

In this section we perform a simple numerical comparison of our Optimal Tensor Method (Algorithm 4) with the Near-Optimal Tensor Method of Gasnikov et al. (2019a); Jiang et al. (2019); Bubeck et al. (2019) (Algorithm 2). We focus on the second-order case ($p = 2$) because it is already a highly important case. We leave investigating higher-order cases ($p \geq 3$) for future work. The main goal of our experimental comparison is to illustrate the practical benefits of eliminating the binary search procedure.

We perform our experiment with logistic regression for binary classification. That is, our objective function $f(x)$ has the form

$$f(x) = \frac{1}{n} \sum_{i=1}^{n} \log(1 + \exp(-b_i a_i^\top x)), \tag{34}$$

where $a_i \in \mathbb{R}^d$ and $b_i \in \{-1, +1\}$ are data points and labels, and $n$ is the number of data points. In the experiment, we use the *a9a* dataset from the LIBSVM[9] dataset collection. This dataset has $n = 32561$ training samples with $d = 123$ features.

We implement both Algorithm 4 and Algorithm 2 in Python language with the help of JAX API.[10] To perform computation on line 5 of Algorithm 2, we use the variant of the binary search procedure described by Bubeck et al. (2019). To perform the computation of the tensor step (7) in both algorithms, we use a procedure based on the eigenvalue decomposition of the Hessian matrix.

For a fair comparison, we use the same value of parameter $M$ in both algorithms. Furthermore, we choose the parameter $\eta$ in Algorithm 4 in such a way that both Algorithm 2 and Algorithm 4 have almost identical convergence in the number of iterations $k$. It is illustrated in Figure 1. It is not a surprise that both algorithms have very similar convergence curves (with our choice of $\eta$) since they are based on the A-HPE Framework (Algorithm 1).

However, the difference between Algorithms 2 and 4 becomes clear when we compare the number of oracle calls performed at each iteration $k$. It is illustrated in Figure 2 from which we can make the following conclusions:

- One can observe that the number of oracle calls performed at each iteration $k$ by Algorithm 4 slightly oscillates around a constant. This is perfectly aligned with our theory which shows that the *average* number of oracle calls performed at each iteration $k$ by Algorithm 4 is bounded by a constant at all times (see, for instance, inequality (31)).

- At the same time, one can observe that the number of oracle calls performed by the binary search procedure at each iteration $k$ of Algorithm 2 slowly grows with the number of iterations $k$. This is also perfectly aligned with the theory of Bubeck et al. (2019); Jiang et al. (2019); Nesterov (2021b) which suggests that the number of iterations performed by the binary search procedure at each iteration $k$ grows as $\mathcal{O}(\log k)$ (in fact, the blue curve in Figure 2 resembles this logarithmic dependency).

Overall, Figure 2 shows that Algorithm 2 has to perform substantially more oracle calls at each iteration $k$ than Algorithm 4 due to the binary search procedure.

The aforementioned major difference between the algorithms results in a significantly slower convergence of Algorithm 2 in the total number of oracle calls compared to Algorithm 4. It is illustrated in Figure 3 which shows that Algorithm 2 requires approximately 2 times more oracle calls than Algorithm 4 to reach accuracy $\|\nabla f(x)\|^2 \leq 10^{-15}$. This difference becomes even more dramatic if we compare both algorithms in the wall clock time which is shown in Table 2. One can observe that Algorithm 4 converges approximately 4 times faster than Algorithm 2.

In summary, our illustrative experiment shows that the proposed Optimal Tensor Method (Algorithm 4) is indeed a practical algorithm. Moreover, it shows that the necessity to use the binary search procedure by the Near-Optimal Tensor Method (Algorithm 2) results in a significantly slower convergence in

---

[9]The LIBSVM (Chang and Lin, 2011) dataset collection is available at `https://www.csie.ntu.edu.tw/~cjlin/libsvmtools/datasets/`.

[10]JAX API (Bradbury et al., 2018) is available at `https://github.com/google/jax`.

Table 2: Wall clock time taken by both algorithms to reach accuracy $\|\nabla f(x)\|^2 \le 10^{-15}$.

| Algorithm | Near-Optimal Tensor Method (Algorithm 2) | Optimal Tensor Method Algorithm 4 |
|---|---|---|
| **Wall clock time** | 552 sec. | 129 sec. |

both the number of oracle calls and the wall clock time, which is perfectly aligned with the existing theory on the binary search procedure (Bubeck et al., 2019; Jiang et al., 2019; Nesterov, 2021b) and our new theory for our new Algorithm 4.

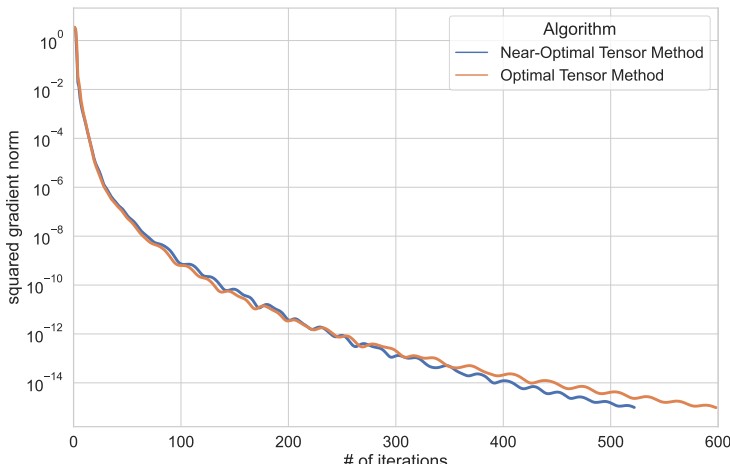

Figure 1: Convergence of the Near-Optimal Tensor Method (Algorithm 2) and Optimal Tensor Method (Algorithm 4) in the number of iterations $k$. We choose parameter $\eta$ of Algorithm 4 in such a way that both algorithms have very similar convergence curves. It is possible because both algorithms are based on the A-HPE framework (Algorithm 1).

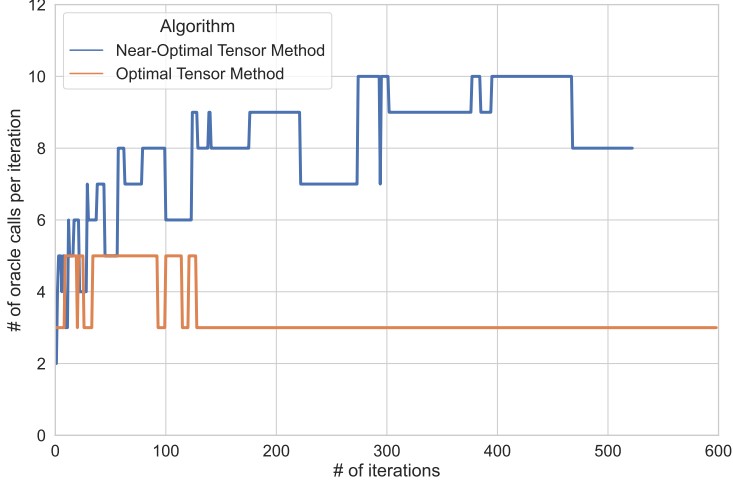

Figure 2: The number of oracle calls performed by the Near-Optimal Tensor Method (Algorithm 2) and Optimal Tensor Method (Algorithm 4) at each iteration $k$. The number of inner iterations of Algorithm 2 grows with the number of iterations $k$ due to the binary search procedure and resembles the logarithmic curve which is perfectly aligned with the existing theory. The number of inner iterations of Algorithm 4 stays constant (and slightly oscillates) which is perfectly aligned with our theory.

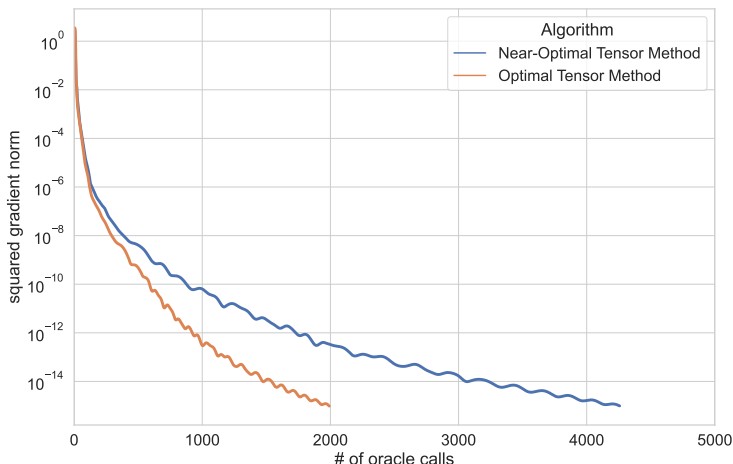

Figure 3: Convergence of the Near-Optimal Tensor Method (Algorithm 2) and Optimal Tensor Method (Algorithm 4) in the total number of oracle calls. Algorithm 2 requires significantly more number of oracle calls to reach a certain precision than Algorithm 4. It is caused by the binary search procedure used by Algorithm 2.