# OpenReview forum: "The First Optimal Acceleration of High-Order Methods in Smooth Convex Optimization"
_NeurIPS.cc/2022/Conference — NeurIPS 2022 Accept_

### Official Review · Reviewer_U1kW · 2022-07-03

**Rating:** 7
**Confidence:** 3
**Soundness:** 3 good
**Presentation:** 4 excellent
**Contribution:** 3 good

**Summary:**

The paper studies highly smooth convex optimization and provides the first optimal acceleration method under oracle model. A function is said to be p-th order smooth if its p-th order gradient is Lipschitz. The previous best known methods for high-order smooth function finds an $\epsilon$-approximation solution using $O(\log(1/\epsilon)\epsilon^{-2/(3p+1)})$ gradient oracles and follows the Monteiro and Svaiter framework, and a lower bound of $\Omega(\epsilon^{-2/(3p+1})$. The paper closes this gap and provide an algorithm using $O(\epsilon^{-2/(3p+1)})$ gradient oracle calls.

To summarize in very short words, the paper still utilizes the Monteiro and Svaiter framework (as previous work), but instead of performing some complex binary search for a fix point problem as a subroutine, the paper gives the explicitly parameter and an explicit high-order extra gradient for searching the ``middle'' point.



**Questions:**

Some additional reference.

[1] Thinking Inside the Ball: Near-Optimal Minimization of the Maximal Loss. COLT'2021
Yair Carmon, Arun Jambulapati, Yujia Jin and Aaron Sidford

[2] Acceleration with a Ball Optimization Oracle
Yair Carmon, Arun Jambulapati, Yujia Jin, Qijia Jiang, Yin Tat Lee, Aaron Sidford and Kevin Tian

**Strengths And Weaknesses:**

Strength. The paper is very well-written, easy to read and give a nice summary on previous work. On the technical side, the removal of binary search and an explicit method for high order smooth optimization is very nice and could be beneficial for practical implementation.


Weakness. There is no major weakness, but personally I think providing some experiments for the methods could make the paper more promising. Though I found the pure theoretical result is already very interesting.

In summary, I think this is a solid paper and could benefit the community. I would suggest to accept the paper.

--------------------------- Post rebuttal ---------------------------

I have read the author's response keep my evaluation, and I vote for acceptance of the paper.

---

> ### Author Response · Authors · 2022-07-31
> **Response to Reviewer U1kW**
>
> We thank Reviewer U1kW for the time and effort. We are glad that the reviewer described our paper as very well-written and easy to read and described our results as interesting and beneficial for practical implementation.
>
> Although Reviewer U1kW did not find any major weaknesses in our paper, Reviewer U1kW suggested that providing some experiments for the methods could make the paper more promising:
>
> >There is no major weakness, but personally I think providing some experiments for the methods could make the paper more promising. Though I found the pure theoretical result is already very interesting.
>
>  Due to this reason, and to properly address the questions raised by other reviewers, we decided to add an experimental comparison of our new algorithm with the existing near-optimal algorithms (Gasnikov et al., Bubeck et al., Jiang et al.). Therefore, we kindly ask Reviewer U1kW to take a look at Appendix F for details.
>
> >Some additional reference.\
> >   [1] Thinking Inside the Ball: Near-Optimal Minimization of the Maximal Loss. COLT'2021 Yair Carmon, Arun Jambulapati, Yujia Jin and Aaron Sidford\
> >   [2] Acceleration with a Ball Optimization Oracle Yair Carmon, Arun Jambulapati, Yujia Jin, Qijia Jiang, Yin Tat Lee, Aaron Sidford and Kevin Tian
>
> We thank the reviewer for pointing us to these works. We will add references to these works and other works that we found after submitting our paper to NeurIPS 2022 in the final version of the paper.
>
> Overall, we hope that our new experimental results will indeed make our paper more promising, as suggested by Reviewer U1kW.

---

> ### Author Response · Authors · 2022-08-07
> **Post rebuttal reply.**
>
> Dear Reviewer U1kW,
> Thank you for your post rebuttal reply. Your feedback is highly appreciated.

---

### Official Review · Reviewer_9tdw · 2022-07-10

**Rating:** 6
**Confidence:** 2
**Soundness:** 3 good
**Presentation:** 3 good
**Contribution:** 3 good

**Summary:**

In this work, the authors propose a p-order method for optimizing a p-times continuously differentiable objective. They prove that the convergence rate matches the theoretical lower bound and is better than the previous works.

**Questions:**

See the weakness points.

**Strengths And Weaknesses:**

Strength:
1. Designing a high-order optimization method that matches the theoretical lower bound on the iteration complexity is indeed a contribution.
2. The theoretical results are complete and solid.

Weakness:
In the abstract, it seems that one of the motivations is that the previously proposed high-order methods are not practical. But in this paper, there are still no numerical experiments. Can you provide some experiments to verify that the proposed method is practical?

---

> ### Author Response · Authors · 2022-07-31
> **Reviewer 9tdw**
>
> We thank Reviewer 9tdw for the time and effort. We are glad that the reviewer found our theoretical results complete and solid. As far as we understand, the only concern raised by the reviewer is the lack of experiments:
>
> > In the abstract, it seems that one of the motivations is that the previously proposed high-order methods are not practical. But in this paper, there are still no numerical experiments. Can you provide some experiments to verify that the proposed method is practical?
>
>  During the rebuttal period, we managed to perform an illustrative experiment that compares our optimal algorithm with the existing near-optimal algorithm (Gasnikov et al., Bubeck et al., Jiang et al.). Our experiment not only shows that our new algorithm is practical but also shows the substantial practical benefits of removing the binary search procedure, which is perfectly aligned with the existing theory and our new theory. We kindly ask the reviewer to take a look at Appendix F for more details.
>
> Overall, we hope that we properly addressed the main concern of Reviewer 9tdw by providing the experiment in Appendix F. Hence, we kindly ask Reviewer 9tdw to consider the possibility of increasing the score if Reviewer 9tdw finds our experimental results convincing.

---

### Official Review · Reviewer_Z7D2 · 2022-07-11

**Rating:** 7
**Confidence:** 4
**Soundness:** 4 excellent
**Presentation:** 2 fair
**Contribution:** 4 excellent

**Summary:**

In this work, the authors propose a new accelerated high-order method for convex optimization. They prove that the global complexity of their method matches the lower complexity bound, thus the method is *optimal*. In the previous approaches, it was required to do some extra line search at each iteration, solving a one-dimensioanal nonlinear equation, which resulted in much more sophisticated algorithms with additional logarithmic factors. In the new method, the line search is not needed. Therefore, this is the first method having the pure *optimal* rate without extra logarithms.

**Questions:**

1. Could you, please, make the difference between Algorithm 1 and 2 more transparent? I found no information in the text why the second algorithm is needed (it seems to be the same as the first one).

2. Line 171: 'the crucial mistake' -- I would propose to use slightly more polite wording, since there is no evidence of mistakes (the old algorithms still have some benefits as it seems, e.g. no need to know R).

3. I think it worths to add some references regarding Tensor Extragradient Method and high-order methods minimizing gradient norm.

**Limitations:**

--

**Strengths And Weaknesses:**

The contribution is the new optimal accelerated high-order method and its analysys. I found these results interesting and significant. To the best of my knowledge, this is the first accelerated (high-order, p >= 2) method that does not use any line search, and matches the lower bounds without extra logarithms.

The main idea of their construction is to use the standard accelerated framework (from the works of Nesterov, 1983, 2018; Monteiro-Svaiter, 2013, Gasnikov et al. 2019) but with a predefined sequence of parameters which provides the method with a required rate of convergence (knowing in advance the optimal rate). Then, to ensure an actual convergence, they propose to solve a prox-point type subproblem by iterations of the basic high-order method (in their case, Tensor Extragradient Method). The main novelty of the analysys is to prove:
1. the convergence of the inner method to a reasonable stationary point in a certain amount of iterations.
2. the boundedness of the total number of inner steps.
While in the previous near-optimal high-order methods, it was required to do high-order step just once, but with solving a complicated nonlinear equation alongside.

In my opinion, this is a solid result which must be of interest for the optimization community.


The weaknesses of the paper are a little bit technical presentation and the lack of any practical evidence (experiments or at least a discussion).
The paper containts a lot of details comparing different analysis techniques with some unnecessary redundancy (e.g. Algorithm 1 and Algorithm 2), however almost no information about how to use their methods in practice. In particular, the following questions seem to be unanswered:

1. How to choose parameter $\nu$ in practice. The formula (30) contains both the Lipschitz constant and the radius R. Is it possible to use some line search? Is it possible to eliminate some of these parameters?

2. How many steps of the inner method are needed in practice, and what happens in case p = 1 (first-order accelerated methods)?

--------------------
After rebuttal:
--------------------
I am happy to increase my score, since I believe that the main theoretical contribution of the paper is very significant for the optimization community.

---

> ### Author Response · Authors · 2022-07-31
> **Response to Reviewer Z7D2**
>
> We thank Reviewer Z7D2 for the time and effort. We are glad that the reviewer found our theoretical results interesting and significant. Further, we provide answers to all the questions raised by the reviewer.
>
> >The weaknesses of the paper are a little bit technical presentation and the lack of any practical evidence (experiments or at least a discussion). The paper containts a lot of details comparing different analysis techniques with some unnecessary redundancy (e.g. Algorithm 1 and Algorithm 2), however almost no information about how to use their methods in practice.
>
> We agree that our paper provides almost no information about practical performance. To address this question, we decided to provide an experimental comparison of our optimal algorithm with the existing near-optimal algorithms (Gasnikov et al., Bubeck et al., Jiang et al.). We kindly ask Reviewer Z7D2 to take a look at Appendix F for more details.
>
> > How to choose parameter $\eta$ in practice. The formula (30) contains both the Lipschitz constant and the radius $R$. Is it possible to use some line search? Is it possible to eliminate some of these parameters?
>
> Parameter $\eta$ can be chosen with a grid search in the same way as stepsizes of various first-order algorithms are chosen in practice. One can choose a maximal value of $\eta$ such that Algorithm 4 performs $T^k \leq 3$ inner iterations of Tensor Extragradient at each iteration $k$. According to our experiment (Appendix F, Figure 2), $T^k$ is approximately a constant throughout all iterations of Algorithm 4. Hence, such a grid search will be cheap because a small number of iterations $k$ is required to see if $T^k \leq 3$.
>
>
> > How many steps of the inner method are needed in practice.
>
> Our theory implies that with a proper choice of $\eta$, the **average** number of inner iterations is bounded as $\mathcal{O}(1)$ (see inequality (31)). Moreover, our experiment (Appendix F, Figure 2) shows that with a proper choice of $\eta$, the number of inner iterations is bounded as $\mathcal{O}(1)$ throughout **all** the iterations of Algorithm 4.
>
> > what happens in case $p = 1$ (first-order accelerated methods)?
>
> Although we don't discuss this in our paper, it is not hard to show that in the case $p=1$ and with $\eta \sim 1/L_1$, the number of inner iterations will be bounded for all $k$: $T^k = \mathcal{O}(1)$. In this case, our algorithm recovers the optimal convergence rate of Accelerated Gradient Descent of Yurii Nesterov.
>
> > Could you, please, make the difference between Algorithm 1 and 2 more transparent? I found no information in the text why the second algorithm is needed (it seems to be the same as the first one).
>
> Algorithm 2 is a special case of Algorithm 1 (just like Algorithm 4 is a special case of Algorithm 1). We added Algorithm 2 for educational purposes to illustrate the issues with this algorithm and why a binary search procedure is needed. We hope that it will help us not only to present new results but to explain the existing results better, making the paper accessible to a broader audience.
>
> > Line 171: 'the crucial mistake' -- I would propose to use slightly more polite wording, since there is no evidence of mistakes (the old algorithms still have some benefits as it seems, e.g. no need to know R).
>
> We apologize for the wording we chose. We did not want to offend the authors of the existing results. We will rewrite this phrase in the final version of the paper.
>
> > I think it worths to add some references regarding Tensor Extragradient Method and high-order methods minimizing gradient norm.
>
> We thank Reviewer Z7D2 for this comment. We agree and will add corresponding references in the final version of the paper, including recent works that we found after submitting our paper to NeurIPS 2022.
>
> ## Conclusion
>
> Overall, we hope that we properly addressed the concerns raised by Reviewer Z7D2 by providing the comments above and illustrative experiment in Appendix F. We kindly ask Reviewer Z7D2 to consider the possibility of increasing the score if Reviewer Z7D2 finds our response and experiment convincing.

---

> > ### Comment · Reviewer_Z7D2 · 2022-08-09
> > **re: Response**
> >
> > Thank you very much for your answers and for providing an illustrative experiment.
> > Please, consider adding your clarifications to the final version of your work as it seems to be helpful for a reader.
> >
> > Overall, after going through the paper and the rebuttal once again, I keep believing that the theoretical contribution of this paper is quite interesting and solid, as it provides us with a new approach to constructing optimal accelerated methods. The paper contains significantly new ideas (which are not some small random tweaks to the previous algorithms, but principally new way of designing and analysing the accelerated high-order methods that leads to the optimal rates).
> >
> > Therefore, I vote for accepting this paper.

---

> > > ### Author Response · Authors · 2022-08-09
> > > **Thank you**
> > >
> > > Dear Reviewer Z7D2,
> > > Thank you for your feedback and increased score. We are glad you appreciated our results and answers to your questions. We, of course, will add the clarifications to the final version of the paper based on the discussions we provided, as it will improve the quality of the paper.

---

### Official Review · Reviewer_drXo · 2022-07-12

**Rating:** 4
**Confidence:** 3
**Soundness:** 4 excellent
**Presentation:** 2 fair
**Contribution:** 2 fair

**Summary:**

This paper studies the question of minimization of convex functions with $p$-order smooth derivatives, given access to an oracle that can minimize the regularized $p$-th order Taylor expansion of the function at a given point.

Recent works (Gasnikov et al, Bubeck et al., Jiang et al.) had previously established an oracle complexity of $O(\epsilon^{-2/(3p+1)}\log 1/\epsilon)$ extending the rate of $O(\epsilon^{-1/2})$ by Accelerated Gradient descent (Nesterov, 1983) for $p=1,$ $O(\epsilon^{-2/7})$ by Montero-Svaiter (2013) for $p=2,$ and Nesterov's accelerated tensor method (2021a) that achieved $O(\epsilon^{-1/(p+1)}).$

A lower bound of $\Omega(\epsilon^{-2/(3p+1)})$ is known from Arjevani et al. (2019). Thus, the known rates were off by a logarithmic factor.

This paper gives an algorithm that achieves a tight bound of $O(\epsilon^{-2/(3p+1)})$ calls to the $p$-th order oracle, thus settling the complexity of $p$-th order smooth convex minimization.

**Questions:**

Am I correct in pointing out that your algorithm does not provide an efficient procedure to compute the next iterate on line 11 in Algorithm 4? If yes, I strongly suggest reflecting this in the definition of Oracle complexity. The current definition of oracle complexity in the paper (Definition 2) only assumes that the oracle returns the higher order derivatives.

**Limitations:**

I have listed the limitations above under weaknesses of the result.

**Strengths And Weaknesses:**

**Strengths**:
The key strength of the result is immediate since it establishes the best-possible oracle complexity of solving $p$-th order smooth convex problems.

**Weakness**:
1. My primary concern is whether shaving off a log factor from the oracle complexity is that significant a question after all. The 3 independent papers that first achieved the almost-optimal complexity had achieved a significant improvement in the oracle complexity. While it is nice to achieve the exactly optimal bound in terms of epsilon, the logarithmic factor seems inconsequential in my opinion.
2. All these papers rely on making calls to an oracle that can not only compute $p$-th order derivatives, but can solve a regularized $p$-th order problem, see Algorithm 1, step 5, eq 15 and Algorithm 3, step 6 or Algorithm 4, line 11. (To the best of my knowledge) We do not know how to solve these problems efficiently for $p > 2$ even given access to the higher order derivatives. In my view, this hides the real complexity of being able to solve highly smooth problems.
3. The notation of the paper is difficult to parse. e.g. in Algorithm 4, $x_g$ is a funny notation that gives no hint to what it means. It relies on $x_f$ which is again not so easy to see where it's defined. In line 11 in Algorithm 4, the notation for $\Phi$ has three(!!) orders of subscript.

---

> ### Author Response · Authors · 2022-07-30
> **Response to Reviewer drXo**
>
> We thank Reviewer drXo for the time and effort. As far as we understand, Reviewer drXo has the following three main concerns regarding our paper:
>
> 1. > My primary concern is whether shaving off a log factor from the oracle complexity is that significant a question after all. The 3 independent papers that first achieved the almost-optimal complexity had achieved a significant improvement in the oracle complexity. While it is nice to achieve the exactly optimal bound in terms of epsilon, the logarithmic factor seems inconsequential in my opinion.
> 2. > All these papers rely on making calls to an oracle that can not only compute $p$-th order derivatives, but can solve a regularized $p$-th order problem, see Algorithm 1, step 5, eq 15 and Algorithm 3, step 6 or Algorithm 4, line 11. (To the best of my knowledge) We do not know how to solve these problems efficiently for $p > 2$ even given access to the higher order derivatives. In my view, this hides the real complexity of being able to solve highly smooth problems.
> 3. > The notation of the paper is difficult to parse. e.g. in Algorithm 4, $x_g$ is a funny notation that gives no hint to what it means. It relies on $x_f$ which is again not so easy to see where it's defined. In line 11 in Algorithm 4, the notation for $\Phi$ has three(!!) orders of subscript.
>
> Further, we address the concerns raised by the reviewer. We will do so in separate posts as we run out of the character limit.

---

> ### Author Response · Authors · 2022-07-30
> **Response to Reviewer drXo (concern 1)**
>
> We respectfully disagree with the statement that the complexity improvement that we provide in our paper is insignificant. Further, we provide strong evidence that the open question we solved (lines 51-52 of our paper) is important.
>
> 1. Let us start with the most fundamental example of smooth convex minimization problems. The lower complexity bound was described by Nemirovsky and Yudin in their seminal book [1]. At that moment, this lower bound was achieved by a variant of the Conjugate Gradient Method (CGM) up to a logarithmic factor only, which is also described in [1]. The logarithmic factor in the complexity appeared due to the necessity to solve an auxiliary 2-dimensional minimization problem at each iteration of CGM, which can be seen as an analog of a line search procedure that caused the logarithmic factor in the near-optimal tensor methods of Gasnikov et al., Bubeck et al., Jiang et al. Hence, the problem of "shaving off" the logarithmic factor in the smooth convex minimization case appeared 40 years ago, and internationally recognized experts in convex optimization tried to solve this problem:
>     - Arkadi Nemirovsky: he improved CGM by replacing a 3-dimensional auxiliary minimization problem with a 2-dimensional problem and later with a 1-dimensional problem [7]. However, he could not reach the lower bound.
>     - Boris Polyak: he worked on his Heavy-ball method, which achieves optimal convergence rate in the quadratic minimization case but can diverge in the general case.
>     - Yurii Nesterov: he reached the lower complexity bound by developing the Accelerated Gradient Descent method during his PhD under supervision of Boris Polyak.
>
>     The AGD method of Yurii Nesterov became very popular not only because it eliminates the logarithmic factor but because it has a simple structure without any additional search procedures. This property of AGD allows generalizations to various settings, including stochastic and finite-sum problems (such as [2]), coordinate methods (such as [3]), primal-dual methods, etc. Our optimal tensor method has similar properties: it has a simple structure and does require any additional search procedures. Hence, we believe that our method will be useful not only due to its efficiency but also due to being a promising method with a lot of potential opportunities for extensions.
>
> 2. Another evidence that the problem we solved is important is that a similar problem was recently solved in the case of high-order methods for variational inequalities (VI). In May 2022, two concurrent preprints [4] and [5] appeared on arXiv, that develop line search-free high-order methods for solving smooth monotone VI problems. Both works [4,5] mention the importance of finding such methods. In particular, the authors of [4] state that line search procedure can be computationally prohibitive in practice, and the problem of finding simple high-order regularization methods remains an open and challenging question in the optimization theory. Please note that these papers appeared on arXiv less than two weeks before the NeurIPS 2022 submission deadline, and we were unaware of them. However, the fact that such works were submitted within a short period by at least two independent research groups suggests that the problem of finding line search-free high-order optimization methods is hot even in the case of VIs, not to mention more popular minimization problems.
>
> 3. Finally, we would like to mention that the problem of finding simple line search-free high-order methods for minimization problems was formulated by Yurii Nesterov, who is a world-recognized expert in convex optimization in general and in high-order optimization in particular. In his book [6, page 305], he noted the difficulty of removing the binary search procedure without sacrificing the rate of convergence and highlighted this as an open and challenging question (the authors of [4] also referred to this). From our private discussions with Yurii Nesterov we found out that he has been trying to solve this problem for 4 years but has not succeeded.
>
> We hope that the three points made above will convince Reviewer drXo that our contribution is significant and has the potential to make a noticeable impact on the community.
>
> [1] Nemirovsky A. S., Yudin D. B. (1983). Problem complexity and method efficiency in optimization. .
>
> [2] Allen-Zhu, Z. (2017). Katyusha: The first direct acceleration of stochastic gradient methods.
>
> [3] Fercoq, O., Richtarik, P. (2015). Accelerated, parallel, and proximal coordinate descent.
>
> [4] Lin, T., Jordan, M. I. (2022). Perseus: A Simple and Optimal High-Order Method for Variational Inequalities.
>
> [5] Adil, D., Bullins, B., Jambulapati, A., Sachdeva, S. (2022). Line Search-Free Methods for Higher-Order Smooth Monotone Variational Inequalities.
>
> [6] Nesterov, Y. (2018). Lectures on convex optimization
>
> [7] A. S. Nemirovsky (1982). Orth-method for smooth convex optimization (in Russian).

---

> > ### Comment · Reviewer_drXo · 2022-08-09
> > **Significance of shaving off the log factors**
> >
> > Thank you for your detailed comments. I appreciate you presenting to me multiple lines of arguments that this problem is significant.
> >
> > I do think it's interesting that you can achieve a guarantee that's exactly optimal. Unfortunately, I respectfully disagree with key idea that shaving off a log(1/epsilon) is significant, and it is a theoretical curiosity in my view.
> >
> > Looking at the results your experiments in the supplementary, Figure 3 demonstrates the number of oracle calls are reduced just by a factor of 2 for the same target error (Figure 2 indicates that it's a factor of 3 if you were to count the number of calls per iteration). This is true even when your chosen target error is $\epsilon = 10^{-14},$ which is basically machine precision, and orders of magnitude smaller than what most practical optimization problems are solved for. (As a comparison, linear systems are typically solved to an error of $10^{-8}.$)

---

> ### Author Response · Authors · 2022-07-30
> **Response to Reviewer drXo (concern 2)**
>
> > All these papers rely on making calls to an oracle that can not only compute $p$-th order derivatives, but can solve a regularized $p$-th order problem, see Algorithm 1, step 5, eq 15 and Algorithm 3, step 6 or Algorithm 4, line 11. (To the best of my knowledge) We do not know how to solve these problems efficiently for $p > 2$ even given access to the higher order derivatives. In my view, this hides the real complexity of being able to solve highly smooth problems.
>
> 1. Please note that the notion of oracle we use is standard and widely used by the optimization community. In particular, this notion of oracle is used in all relevant papers that we are aware of, including (Nesterov, 2021a and 2021b), (Agarwal and Hazan, 2018), (Arjevani et al., 2019), (Jiang et al., 2019), etc. Hence, we can not change our oracle definition as it would contradict the standards of the community.
> 2. We respectfully disagree that we don't know how to solve the auxiliary subproblems efficiently in the case $p>2$. Yurii Nesterov provided an efficient procedure for solving such problems in the case $p = 3$ (Nesterov, 2021a), which is based on the optimization framework of relatively smooth functions.
>
> Overall, our result is already highly important even in the case $p=2$, because we provide the first optimal second-order optimization algorithm. In the case $p=3$, an efficient procedure for solving the auxiliary subproblem is developed by Yurii Nesterov in his work on implementable tensor methods (Nesterov, 2021a), which makes our algorithm implementable as well.
>
> On the other hand, the concern raised can be applied to any existing paper on the topic of high-order optimization. While we justified that efficient implementation of our algorithm is possible, at least in the case $p=2$ and $p=3$, we kindly ask to note that the goal of our paper is to provide an optimal acceleration of high-order methods (which is already a highly important problem as we discussed in our response to concern 1) rather than to develop efficient procedures for solving the auxiliary subproblems. At the same time, we will be happy to provide a brief discussion of the aforementioned implementation details in the revised version of our paper, as it will likely improve the paper. We thank Reviewer drXo for drawing our attention to this.

---

> > ### Comment · Reviewer_drXo · 2022-08-09
> > **Higher order oracle**
> >
> > Thank you for pointing out that the oracle can be solved in the $p=3$ case too, but that does little to address larger $p.$ Please see my next response too.
> >
> > I am aware that this is the same oracle that has been used previously in highly smooth convex optimization results.

---

> ### Author Response · Authors · 2022-07-30
> **Response to Reviewer drXo (concern 3)**
>
> > The notation of the paper is difficult to parse. e.g. in Algorithm 4, $x_g$ is a funny notation that gives no hint to what it means. It relies on $x_f$ which is again not so easy to see where it's defined. In line 11 in Algorithm 4, the notation for $\Phi$ has three(!!) orders of subscript.
>
> With all due respect, we can not understand what is funny about our notation. We kindly ask Reviewer drXo to note that the notation clarity is somewhat subjective. For instance, our notation is motivated by the Accelerated Gradient Descent method, and we find it very intuitive, especially for the purpose of writing mathematical proofs: $x_g^k$ has the subscript **g** because it is the point from which the **g**radient step is taken in the AGD method, $x_f^k$ has the subscript **f** because it is the point at which the **f**unction value $f(x_f^k)$ is computed in the proof of the AGD method. We think having such a sequence of points $(x^k,x_g^k,x_f^k)$ is more intuitive than having a sequence with different letters such as $(x^k,y^k,z^k)$ or $(v^k,y^k,x^k)$ in which it is much easier to confuse the meaning of the letters.
>
> Unfortunately, high-order methods are typically more complicated than the first-order methods. Hence, if we want to write the iterations of our algorithm in a compact way we have to use many subscripts or many function arguments. For instance, similar issue can be seen in Gasnikov et al. (Algorithm 1, line 2):
> $$
> y^{k+1} = T_{p,p M_p}^{F_{L_k,x_k}}(x^k).
> $$
> Regarding, the line 11 of Algortihm 4, we can rewrite this line in the following way:
> $$
> A^k(\cdot) =  A_{\lambda_{k}}(\cdot;x_g^k),
> $$
> $$
> x^{k,t+1/2} = \arg\min_{x \in \mathbb{R}^d} \Phi_{A^k(\cdot)}^p (x;x^{k,t}) + \frac{pM}{(p+1)!}\||x - x^{k,t}\||^{p+1}.
> $$
> In this case, $\Phi$ will have less subscripts.
>
> Overall, we kindly ask Reviewer drXo to note that different readers may prefer different types of notation and we can't satisfy everyone. On the other hand, the notation issues seem to be minor and do not influence the quality of our results.

---

> > ### Comment · Reviewer_drXo · 2022-08-09
> > **Notation**
> >
> > Thank you for explaining your notation, this was helpful and not obvious from the paper. I hope you will add it to your revision.
> >
> > Arguing that other papers have ugly notation doesn't help me understand your notation.
> >
> > The hope is good notation has enough hints to help the reader remind them of their meaning. e.g. The way you explained why you named the points $x^g$ and $x^f,$ maybe it's better to name the Taylor polynomial with $T$ or $\mathcal{T}$ rather than $\Phi.$ As another example, the definition of $A$ is maybe unnecessary, and you could just spell it out explicitly in your claims. e.g. in (16), the Taylor polynomial of $A_{\lambda_k}$ could just become the Taylor polynomial of $f$ plus the quadratic regularization term from (13).
> >
> > (I should say that the notation issue is minor, and did not affect my evaluation of the paper)

---

> ### Author Response · Authors · 2022-07-31
> **Response to Reviewer drXo (question and conclusion)**
>
> Reviewer drXo asked the following question that we address here:
>
> > Am I correct in pointing out that your algorithm does not provide an efficient procedure to compute the next iterate on line 11 in Algorithm 4? If yes, I strongly suggest reflecting this in the definition of Oracle complexity. The current definition of oracle complexity in the paper (Definition 2) only assumes that the oracle returns the higher order derivatives.
>
> The answer to this question is contained in our response to the 2nd concern. In summary, efficient procedures for solving the auxiliary subproblem exist in the literature at least in the case $p=2$ and $p=3$, and we will be happy to provide a brief description of such procedures in the revised version of our paper. However, searching for such procedures in the case $p>3$ is an open issue which is beyond the scope of our work (finding an optimal acceleration of high-order optimization methods) and can be applied to any paper on the topic of high-order optimization.
>
> ## Conclusion
>
> We are convinced that we addressed the concerns raised by Reviewer drXo, including the primary concern about the significance of our result:
> - We justified the importance of exact achieving the lower complexity bounds by providing a relevant historical example of how a similar development resulted in one of the most significant breakthroughs in recent optimization history.
> - We also showed that the problem we solved is hot since a very similar problem has been solved recently by at least two independent research groups in the neighboring field of high-order variational inequalities (VI). However, the high-order VI problems are much simpler than the problem we consider because VIs do not require acceleration and the lower bounds are much easier to achieve in the case of high-order VIs.
> - Finally, during the rebuttal period, we managed to perform an experimental comparison of our new algorithm with the existing near-optimal algorithms. Our experiment shows that removing the binary search procedure, even in the case $p=2$, results in a substantial speed-up both in the number of oracle calls and wall clock time. This shows that the problem we solved is significant not only from the theoretical point of view but also from the practical perspective. We kindly ask Reviewer drXo to take a look at Appendix F for experimental details.
>
> We kindly ask Reviewer drXo to reconsider the decision towards the acceptance or provide additional feedback so that we could better address the concerns.

---

> > ### Comment · Reviewer_drXo · 2022-08-09
> > **Response**
> >
> > I really appreciate the time and effort you have taken to address my questions, and conduct experimental comparisons.
> >
> > Please see my responses to your individual comments below. The notation issue is minor. The summary of my stand is that theoretically your method improves on log factors, and your experiments demonstrate that the oracle calls are improved by a factor of 2-3. This is not significant in comparison to the fact that the complexity of the oracle call is hidden (and so it is in other higher-order smoothness papers). I regret that I cannot support the paper.

---

> > > ### Author Response · Authors · 2022-08-09
> > > **Final comments (1/2)**
> > >
> > > Dear Reviewer drXo,
> > >
> > > Thank you for your feedback. Further, we provide additional comments.
> > >
> > > ## Significance
> > > >I respectfully disagree with key idea that shaving off a log(1/epsilon) is significant.
> > >
> > > Unfortunately, you do not provide any evidence of this claim whatsoever.
> > > In contrast, we provide a highly detailed argumentation that supports the exact opposite statement: **removing the logarithmic factors from the complexity and achieving lower complexity bounds is a significant contribution**. If you disagree with our arguments, you should provide strong evidence that our justification is incorrect. The only comment about our justification we could find was
> > >
> > > > I appreciate you presenting to me multiple lines of arguments that this problem is significant.
> > >
> > > Unfortunately, this comment does not give any clue why the provided justification is incorrect.
> > > In addition, we would like to elaborate further on the significance of the problem. Removing logarithmic factors in complexities and reaching optimal rates have been studied by researchers for years, including:
> > >
> > > - Zeyuan Allen-Zhu and Elad Hazan developed an optimal regularization technique in their NeurIPS 2016 paper *"Optimal black-box reductions between optimization objectives."* They state the contribution of removing the $\log 1/\epsilon$ factor from the existing regularization techniques as their main contribution. (they even use the same wording: "shave off"). They argue that "obtaining the optimal convergence rate is one of the main goals in operations research and machine learning."
> > >
> > > - Another major breakthrough was obtaining the optimal $1/\epsilon$ rate for online learning, compared to the existing rate $\log (1/\epsilon) /\epsilon$. It was done by three (!) independent and respected research groups: *"Beyond the regret minimization barrier: Optimal algorithms for stochastic strongly-convex optimization"* (JMLR 2014) by Elad Hazan and Satyen Kale, *"A simpler approach to obtaining an $o(1/t)$ convergence rate for the projected stochastic subgradient method."* by Simon Lacoste-Julien, Mark Schmidt, and Francis Bach, *"Making gradient descent optimal for strongly convex stochastic optimization"* (ICML 2012) by Alexander Rakhlin, Ohad Shamir, and Karthik Sridharan.
> > >
> > > - Another classic example is the paper *"Minimax policies for adversarial and stochastic bandits"* (COLT 2009) by Jean-Yves Audibert and Sebastien Bubeck, who resolved a long-standing open problem of removing the logarithmic factor in the upper bound for the multi-armed bandit problem.

---

> > > ### Author Response · Authors · 2022-08-09
> > > **Final comments (2/2)**
> > >
> > > - The next example is finding an optimal primal algorithm for solving decentralized distributed optimization problems over fixed networks. We can mention at least five independent works that tried to achieve the optimal convergence rate but, for some reasons, failed and ended up with extra logarithmic factors:
> > >    1. *"A sharp convergence rate analysis for distributed accelerated gradient methods"* by Huan Li, Cong Fang, Wotao Yin, and Zhouchen Lin;
> > >    2. *"Decentralized and parallel primal and dual accelerated methods for stochastic convex programming problems."* by Darina Dvinskikh and Alexander Gasnikov;
> > >    3. *"A dual approach for optimal algorithms in distributed optimization over networks"* by Cesar Uribe, Soomin Lee, Alexander Gasnikov, and Angelia Nedic;
> > >    4. *"Multi-consensus decentralized accelerated gradient descent"* by Haishan Ye, Luo Luo, Ziang Zhou, and Tong Zhang;
> > >    5. *"Ideal: Inexact decentralized accelerated augmented lagrangian method"* (NeurIPS 2020) by Yossi Arjevani, Joan Bruna, Bugra Can, Mert Gurbuzbalaban, Stefanie Jegelka, and Hongzhou Lin.
> > >
> > >     This problem of removing the logarithmic factor in decentralized distributed optimization was solved only recently by Dmitry Kovalev, Adil Salim, and Peter Richtarik *"Optimal and practical algorithms for smooth and strongly convex decentralized optimization"* (NeurIPS 2020).
> > >
> > > - The problem of removing the logarithm factor in optimal variance reduced algorithms for solving finite-sum optimization problems was solved independently in the paper *"Katyusha: The first direct acceleration of stochastic gradient methods"* (JMLR 2017) by Zeyuan Alle-Zhu and in the paper *"An optimal randomized incremental gradient method"* (Mathematical Programming) by George Lan and Yi Zhou (730 citations since 2015).
> > >
> > > - The logarithmic factor was removed in the optimal high-order algorithms for variational inequalities by two respected independent groups (we already described it): Tianyi Lin, Michael Jordan, and Deeksha Adil, Brian Bullins, Arun Jambulapati, Sushant Sachdeva.
> > >
> > > - The logarithmic factor was removed from the CGM method for solving minimization problems 40 years ago by Yurii Nesterov (we already described it).
> > >
> > > One can continue this list of examples for much longer. These are all internationally recognized experts in optimization research who fight to remove those logarithmic factors in order to reach lower complexity bounds in various optimization problems. With all that said, we have to conclude that our position is strongly supported by arguments. Hence, we respectfully reject your criticism of significance because it is **invalid** and, most importantly, **unjustified**.
> > >
> > >
> > > ## Experiments
> > >
> > > > Looking at the results your experiments in the supplementary, Figure 3 demonstrates the number of oracle calls are reduced just by a factor of 2 for the same target error (Figure 2 indicates that it's a factor of 3 if you were to count the number of calls per iteration). This is true even when your chosen target error is $\epsilon^{-14}$, which is basically machine precision, and orders of magnitude smaller than what most practical optimization problems are solved for. (As a comparison, linear systems are typically solved to an error of $10^{-8}$).
> > >
> > > The plot with the number of oracle calls serves for the following reason: to confirm the theoretical predictions. It perfectly serves this purpose, as it shows that in the existing methods, the number of inner-loop oracle calls grows logarithmically, while this number remains constant in our algorithm.
> > >
> > > On the other hand, we see a contradiction: you argue that the number of oracle calls (with our definition of oracle calls) is not a good measure of efficiency, but you use it as a justification for the insufficient efficiency of our algorithm. At the same time, you ignore the **wall clock time** measurement, which is a perfect indicator of efficiency and is also provided.
> > >
> > > Further, we provide a table that compares the wall clock time required by the old (near optimal) algorithms and the new algorithm to reach precision $10^{-8}$, which is the precision you requested. We compare on 3 datasets. Our algorithm has a **clear and significant advantage** (by 3-4 times) in the requested scenario.
> > >
> > > | dataset  | a9a     | w9a      | madelon  |
> > > | -------- | ------- | -------- | -------- |
> > > | old alg. | 49 sec. | 425 sec. | 384 sec. |
> > > | new alg. | 18 sec. | 99 sec.  | 90 sec.  |
> > >
> > > ## Conclusion
> > >
> > > Our optimal algorithms significantly outperform the existing state-of-the-art algorithms from both theoretical and practical points of view. Our experiments nicely do the following things: they confirm the theory and show the significant practical advantage of our algorithm.
> > >
> > > We kindly ask you to reconsider your score towards acceptance of our paper because we provide strong and clear evidence of the significance of our results from both theoretical and practical perspectives.

---

### Author Response · Authors · 2022-07-31
**Message to all Reviewers (summary and numerical experiment)**

Dear Reviewers,

We thank you for your time and valuable comments on our paper.
A lot of positive was said about our paper, for instance:

**By Reviewer drXo:**
> it establishes the best-possible oracle complexity of solving $p$-th order smooth convex problems.

**By Reviewer Z7D2:**
>  I found these results interesting and significant. To the best of my knowledge, this is the first accelerated (high-order, $p\geq 2$) method that does not use any line search, and matches the lower bounds without extra logarithms.

**By Reviewer 9tdw:**
> Designing a high-order optimization method that matches the theoretical lower bound on the iteration complexity is indeed a contribution. The theoretical results are complete and solid.

**By Reviewer U1kW:**
>  The paper is very well-written, easy to read and give a nice summary on previous work. On the technical side, the removal of binary search and an explicit method for high order smooth optimization is very nice and could be beneficial for practical implementation.


At the same time a few questions were raised by reviewers. We provide answers to these questions and detailed comments on each review in separate posts.

## Illustrative experiments

As far as we understand, a minor concern that the reviewers raised is that we did not provide numerical experiments with our algorithm. We are strongly convinced that our paper offers strong theoretical results and does not require any experiments (just like many excellent experimental papers accepted to NeurIPS do not provide any theory). However, we decided to perform a simple experimental comparison of our new optimal algorithm with the existing near-optimal algorithm (Gasnikov et al., Bubeck et al., Jiang et al.) because it will help us to address some of the questions raised by the reviewers.

Our experiment is provided in Section F of the Appendix. Please note that we have not applied any changes to the main part of the paper suggested by the reviewers due to a lack of time. However, we will be happy to apply those changes in the final version of the paper.

In summary, our experiment is perfectly aligned with the theory. It shows that the line search procedure indeed requires substantially more computations at each iteration of the A-HPE Algorithm compared to our new Algorithm 4. As a result, our optimal algorithm substantially outperforms the existing near-optimal algorithm both in the number of oracle calls and wall clock time. We kindly ask the reviewers to take a look at Appendix F for more details.

---

### Meta-Review · Area_Chair_GhvQ · 2022-08-21

**Recommendation:** Accept
**Confidence:** Less certain

**Metareview:**

The paper is interesting and the reviewers on average recommend (weak-ish) acceptance. I agree with that asessment.

**Award:**

No

---

### Decision · Program_Chairs · 2022-09-14

Accept